# INSTANT POLICY: IN-CONTEXT IMITATION LEARNING VIA GRAPH DIFFUSION

**Vitalis Vosylius and Edward Johns**
The Robot Learning Lab at Imperial College London
`vitalis.vosylius19@imperial.ac.uk`

## ABSTRACT

Following the impressive capabilities of in-context learning with large transformers, In-Context Imitation Learning (ICIL) is a promising opportunity for robotics. We introduce *Instant Policy*, which learns new tasks instantly (without further training) from just one or two demonstrations, achieving ICIL through two key components. First, we introduce inductive biases through a graph representation and model ICIL as a graph generation problem with a learned diffusion process, enabling structured reasoning over demonstrations, observations, and actions. Second, we show that such a model can be trained using pseudo-demonstrations – arbitrary trajectories generated in simulation – as a virtually infinite pool of training data. Simulated and real experiments show that Instant Policy enables rapid learning of various everyday robot tasks. We also show how it can serve as a foundation for cross-embodiment and zero-shot transfer to language-defined tasks. Code and videos are available at https://www.robot-learning.uk/instant-policy.

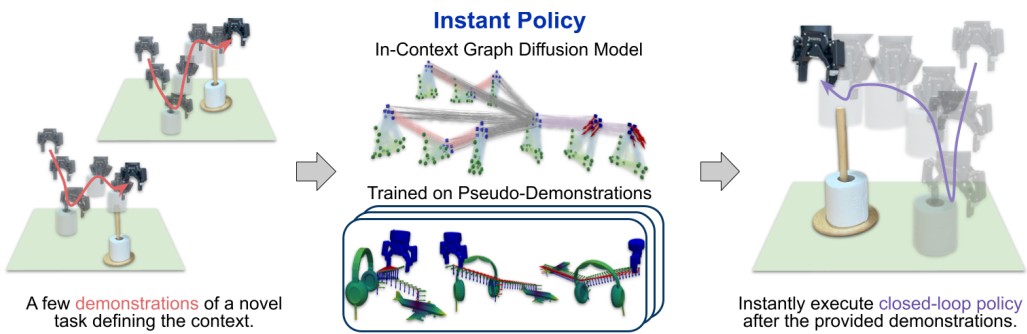

Figure 1: **Instant Policy** acquires skills instantly after providing demos at test time. We model in-context imitation learning as a graph-based diffusion process, trained using pseudo-demonstrations.

## 1 INTRODUCTION

Robot policies acquired through Imitation Learning (IL) have recently shown impressive capabilities, but today's Behavioural Cloning (BC) methods still require hundreds or thousands of demonstrations per task (Zhao et al.). Meanwhile, language and vision communities have shown that when large transformers are trained on sufficiently large and diverse datasets, we see the emergence of In-Context Learning (ICL) (Brown, 2020). Here, trained models can use test-time examples of a novel task (the context), and instantly generalise to new instances of this task without updating the model's weights. This now offers a promising opportunity of In-Context Imitation Learning (ICIL) in robotics. To this end, we present *Instant Policy*, which enables new tasks to be learned instantly: after providing just one or two demonstrations, new configurations of that task can be performed immediately, without any further training. This is far more time-efficient and convenient than BC methods, which require numerous demonstrations and hours of network training for each new task.

ICL in language and vision benefits from huge and readily available datasets, which do not exist for robotics. As such, we are faced with two primary challenges. **1)** Given the limited available data, we need appropriate inductive biases in observation and action representations for efficient learning in

3D space; **2)** Given the inefficiency and cost of manually collecting robotics data, we need a means to easily collect training data in a scalable way. In this work, we propose solutions to both these.

We address the first challenge by introducing a novel graph-based representation that integrates demonstrations, current point cloud observations, and the robot's actions within a unified graph space. We then cast ICIL as a diffusion-based graph generation problem, enabling demonstrations and observations to be interpreted effectively in order to predict the robot's actions. To address the second challenge, we observe that in traditional BC, the model's weights directly encode policies for a specific set of tasks, whereas in ICIL, the model's weights should encode a more general, task-agnostic ability to interpret and act upon the given context. Due to this, we found that we were able to train the model using *pseudo-demonstrations* – sets of procedurally generated robot trajectories, but where each set of demonstrations for a task is semantically consistent. This approach allows us to generate virtually infinite training data by scaling up the simulated data.

Our experiments, with both simulated and real-world tasks, show that Instant Policy can learn various everyday tasks, whilst achieving higher task success rates than state-of-the-art baselines trained on the same data. As an emergent ability, we also observed generalisation capabilities to object geometries unseen from the test-time demonstrations. Importantly, we found that performance improves as more data is generated and used for simultaneous training, offering scalable opportunities with abundant simulated data. In our further experiments on downstream applications, Instant Policy also achieves cross-embodiment transfer from human-hand demonstrations to robot policies, and zero-shot transfer to language-defined tasks without needing large language-annotated datasets.

Our contributions are as follows: **1)** We cast In-Context Imitation Learning as a diffusion-based graph generation problem; **2)** We show that this model can be trained using procedurally generated pseudo-demonstrations; **3)** We evaluate in simulation and the real world across various everyday tasks, showing strong performance, encouraging scaling trends, and promising downstream uses.

## 2 RELATED WORK

**In-Context Learning (ICL).** ICL is an emerging paradigm in machine learning which allows models to adapt to new tasks using a small number of examples, without requiring explicit weight updates or retraining. Initially popularised in natural language processing with models like GPT-3 (Brown, 2020), ICL has been applied to enable robots to rapidly adapt to new tasks by using foundation models (Di Palo & Johns, 2024), finding consistent object alignments (Vosylius & Johns, 2023a), identifying invariant regions of the state space (Zhang & Boularias, 2024), and directly training policies aimed at task generalisation (Duan et al., 2017; Fu et al., 2024) or cross-embodiment transfer (Jang et al., 2022; Jain et al., 2024; Vogt et al., 2017). Despite these advancements, challenges remain in achieving generalisation to tasks unseen during training and novel object geometries. Instant Policy addresses this by leveraging simulated pseudo-demonstrations to generate abundant and diverse data, while its structured graph representation ensures that this data is utilised efficiently.

**Diffusion Models.** Diffusion models (Ho et al., 2020) have garnered significant attention across various domains, due to their ability to iteratively refine randomly sampled noise through a learned denoising process, ultimately generating high-quality samples from the underlying distribution. Initially popularised for image generation (Ramesh et al., 2021), diffusion models have recently been applied to robotics. They have been utilised for creating image augmentations (Yu et al., 2023; Mandlekar et al., 2023) to help robots adapt to diverse environments, generating 'imagined' goals (Kapelyukh et al., 2023) or subgoals (Black et al., 2023) for guiding robotic policies, and learning precise control policies (Chi et al., 2023; Vosylius et al., 2024). In contrast, our work proposes a novel use of diffusion models for graph generation, enabling structured learning of complex distributions.

**Graph Neural Networks (GNNs).** Graph Neural Networks (GNNs) allow learning on structured data using message-passing or attention-based strategies. These capabilities have been applied across a wide range of domains, including molecular chemistry Jha et al. (2022), social network analysis (Hu et al., 2021), and recommendation systems (Shi et al., 2018). In robotics, GNNs have been employed for obtaining reinforcement learning (RL) policies (Wang et al., 2018; Sferrazza et al., 2024), managing object rearrangement tasks (Kapelyukh & Johns, 2022), and learning affordance models for skill transfer Vosylius & Johns (2023b). In our work, we build on these foundations by studying structured graph representations for ICIL, enabling learning of the relationships between demonstrations, observations, and actions.

# 3 INSTANT POLICY

## 3.1 OVERVIEW & PROBLEM FORMULATION

**Overview.** We address the problem of In-Context Imitation Learning, where the goal is for the robot to complete a novel task immediately after the provided demonstrations. At test time, one or a few demos of a novel task are provided to define the context, which our trained Instant Policy network interprets together with the current point cloud observation, and infers robot actions suitable for closed-loop reactive control (see Figure 1). This enables instantaneous policy acquisition, without extensive real-world data collection or training. We achieve this through a structured graph representation (Section 3.2), a learned diffusion process (Section 3.3), and an abundant source of diverse simulated pseudo-demonstrations (Section 3.4).

**Problem Formulation.** We express robot actions $a$ as end-effector displacements $\boldsymbol{T}_{EA} \in \mathbb{SE}(3)$ (which, when time-scaled, correspond to velocities), along with binary open-close commands for the gripper, $a_g \in \{0, 1\}$. Such actions move the robot's gripper from frame $E$ to a new frame $A$ and change its binary state accordingly. Our observations, $o_t$ at timestep $t$, consist of segmented point clouds $P^t$, the current end-effector pose in the world frame $W$, $\boldsymbol{T}_{WE}^t \in \mathbb{SE}(3)$, and a binary gripper state $s_g^t \in \{0, 1\}$. Formally, our goal is to find a probability distribution, $p(\boldsymbol{a}^{t:t+T} \mid o_t, \{(o_{ij}, \boldsymbol{a}_{ij})_{i=1}^L\}_{j=1}^N)$, from which robot actions can be sampled and executed. Here, $T$ denotes the action prediction horizon, while $L$ and $N$ represent the demonstration length and the number of demonstrations, respectively. For conciseness, from now onwards we refer to the demonstrations, which define the task at test time and are not used during training, as context $C$, and the action predictions as $a$. Analytically defining such a distribution is infeasible, therefore we aim to learn it from simulated pseudo-demonstrations using a novel graph-based diffusion process.

## 3.2 GRAPH REPRESENTATION

To learn the described conditional probability of actions, we first need to choose a suitable representation that would capture the key elements of the problem and introduce appropriate inductive biases. We propose a heterogeneous graph that jointly expresses context, current observation, and future actions, capturing complex relationships between the robot and the environment and ensuring that the relevant information is aggregated and propagated in a meaningful manner. This graph is constructed using segmented point cloud observations, as shown in Figure 2.

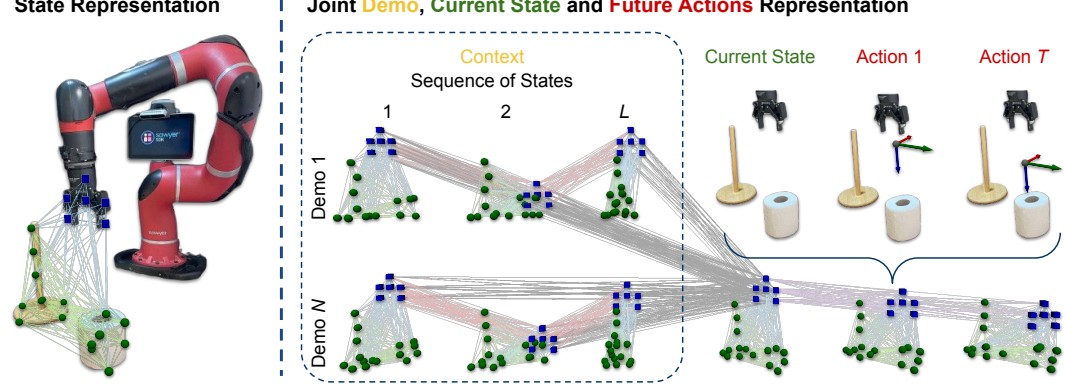

Figure 2: (Left) A local graph, $\mathcal{G}_l$, representing the robot's state (blue nodes) and local geometries of the objects (green nodes). (Right) A graph representing 2 demos (3 waypoints each), the current state, and 2 future actions. Edge colours represent different edge types in a heterogeneous graph.

**Local Representation.** The core building block of our representation is the observation at time step $t$, which we express as a local graph $\mathcal{G}_l^t(P^t, \boldsymbol{T}_{WE}^t, s_g)$ (Figure 2, left). First, we sample $M$ points from a dense point cloud $P^t$ using the Furthest Point Sampling Algorithm and encode local geometry around them with Set Abstraction (SA) layers (Qi et al., 2017), obtaining feature vectors $\mathcal{F}$ and positions $p$ as $\{\mathcal{F}^i, p^i\}_{i=1}^M = \phi(P^t)$. The separately pre-trained $\phi$, an implicit oc-

cupancy network (Mescheder et al., 2019), ensures these features describe local geometries (details in Appendix A). We then represent the gripper's state in the same format, $\{\mathcal{F}_g^i, p_g^i\}_{i=1}^6$, by rigidly transforming key points $p_{kp}$ on the end-effector, $p_g = \boldsymbol{T}_{WE} \times p_{kp}$ and assigning them embeddings that encode node distinction and gripper state information $\mathcal{F}_g^i = [f_g^i, \phi_g(s_g)]^T$. Finally, we link scene and gripper nodes with directional edges and assign edge attributes $e$ representing relative positions in Cartesian space. To increase the precision and capture high-frequency changes in the positions of the described nodes, we represent edges as $e_{ij} = (sin(2^0 \pi(p_j - p_i)), cos(2^0 \pi(p_j - p_i)), ..., sin(2^{D-1} \pi(p_j - p_i)), cos(2^{D-1} \pi(p_j - p_i)))$, similar to Zhou et al. (2023).

**Context Representation.** While $\mathcal{G}_l^t$ captures the environment state, a sequence of such graphs, interleaved with actions, defines a trajectory within context $C$ (Figure 2, middle). We perform this interleaving by linking gripper nodes across time to represent their relative movement (red edges) and connecting all demonstration gripper nodes to the current ones to propagate relevant information (grey edges). This enables the graph to efficiently handle any number of demos, regardless of length, whilst ensuring that the number of edges grows linearly. The result, $\mathcal{G}_c(\mathcal{G}_l^t, \{\mathcal{G}_l^{1:L}\}_1^N)$, enables a structured flow of information between the context and the current observation.

**Action Representation.** To express future actions $\boldsymbol{a} = (\boldsymbol{T}_{EA}, a_g)$ within the graph representation, we construct local graphs as if the actions were executed and the gripper moved: $\mathcal{G}_l^a(P^t, \boldsymbol{T}_{WE}^t \times \boldsymbol{T}_{EA}, a_g)$. This allows 'imagining' spatial implications of actions. Thus, the actions are fully described within the positions and the features of the nodes of these local graphs. To represent actions as relative movements from the current gripper pose, we then add edges between current and future gripper nodes with position-based embeddings that represent relative movement between subsequent timesteps. These edges propagate the information from the current observation (and indirectly the context) to the nodes representing the actions. The final graph, $\mathcal{G}(\mathcal{G}_l^a(\boldsymbol{a}), \mathcal{G}_c(\mathcal{G}_l^t, \{\mathcal{G}_l^{1:L}\}_1^N))$, aggregates relevant information from the context and the current observation and propagates it to nodes representing the actions, enabling effective reasoning about the robot actions by ensuring the observations and actions are expressed in the same graph space.

## 3.3 LEARNING ROBOT ACTION VIA GRAPH DIFFUSION

To utilise our graph representation effectively, we frame ICIL as a graph generation problem and learn a distribution over previously described graphs $p_\theta(\mathcal{G})$ using a diffusion model, depicted in Figure 3. This approach involves forward and backward Markov-chain processes, where the graph is altered and reconstructed in each phase. At test time, the model iteratively updates only the parts of the graph representing robot actions, implicitly modelling the desired conditional action probability.

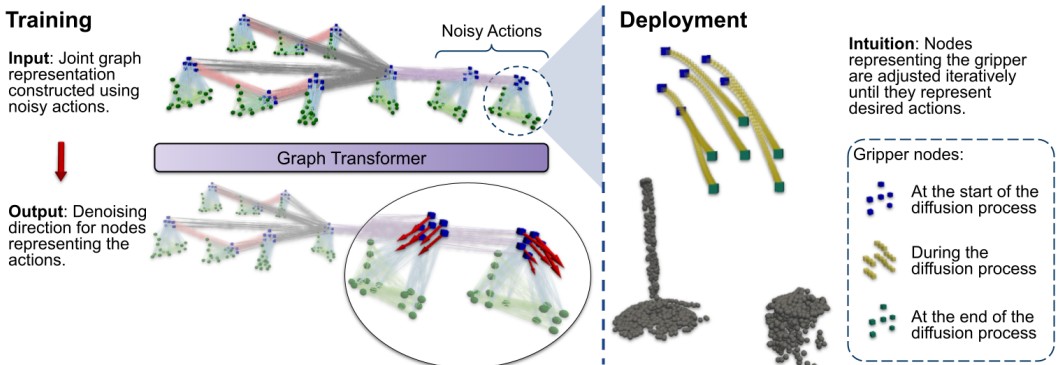

Figure 3: (Left) High-level structure of the network used to train graph-based diffusion model. (Right) Position of gripper action nodes during the denoising process for one of the predicted actions.

**Training.** Training our diffusion model includes, firstly, the forward process, where noise is iteratively added to the samples extracted from the underlying data distribution $q(\mathcal{G})$. In this phase, we construct a noise-altered graph by adding noise to the robot actions according to Ho et al. (2020):

$$q(\mathcal{G}^k \mid \mathcal{G}^{k-1}) = \mathcal{G}(\mathcal{G}_l^a(\mathcal{N}(\boldsymbol{a}^k; \sqrt{1-\beta_k}\boldsymbol{a}^{k-1}, \beta_k \mathbf{I}), \mathcal{G}_c)), \quad k = 1, \dots, K \qquad (1)$$

Here, $\mathcal{N}$ represents the normal distribution, $\beta_k$ the variance schedule, and $K$ the total number of diffusion steps. This process gradually transitions the ground truth graph representation into a graph constructed using actions sampled from a Gaussian distribution.

Inversely, in the reverse diffusion process, the aim is to reconstruct the original data sample, in our case the graph, from its noise-altered state, utilising a parameterised model $p_\theta(\mathcal{G}^{k-1} \mid \mathcal{G}^k)$. Intuitively, such a model needs to learn how the gripper nodes representing the robot actions should be adjusted, such that the whole graph moves closer to the true data distribution $q(\mathcal{G})$. Formally, the parameterised model learns a denoising process of actions using our graph representation $\mathcal{G}(\boldsymbol{a})$ as:

$$\mathcal{G}^{k-1} = \mathcal{G}(\mathcal{G}_l^a(\alpha(\boldsymbol{a}^k - \gamma\varepsilon_\theta(\mathcal{G}^k, k)) + \mathcal{N}(0, \sigma^2\boldsymbol{I})), \mathcal{G}_c) \tag{2}$$

Here, $\varepsilon_\theta(.)$ can be interpreted as effectively predicting the gradient field, based on which a single noisy gradient descent step is taken (Chi et al., 2023). As we represent actions as collections of nodes with their associated positions $p$ and features, that depend on the binary gripper actions $a_g$, such a gradient field has two components $\varepsilon_\theta = [\nabla p, \nabla a_g]^T$. As we will discuss later, $\nabla a_g$ can be used directly in the diffusion process, while a set of $\nabla p$ predictions is an over-parameterisation of a gradient direction on the $\mathbb{SE}(3)$ manifold, and additional steps need to be used to compute a precise denoising update. However, this can result in a large translation dominating a small rotation, and vice versa, preventing precisely learning both components well. To address this, we represent the denoising directions as a combination of centre-of-mass movement and rotation around it, effectively decoupling the translation and rotation predictions while remaining in Cartesian space as $[\nabla\hat{p}_t, \nabla\hat{p}_r]^T = [\boldsymbol{t}_{EA}^0 - \boldsymbol{t}_{EA}^k, \boldsymbol{R}_{EA}^0 \times p_{kp} - \boldsymbol{R}_{EA}^k \times p_{kp}]^T$, with $\nabla\hat{p} = \nabla\hat{p}_t + \nabla\hat{p}_r$ representing flow (red arrows in Figure 3, left). Here, $\boldsymbol{t}_{EA} \in \mathbb{R}^3$ and $\boldsymbol{R}_{EA} \in \mathbb{SO}(3)$ define the $\mathbb{SE}(3)$ transformation representing actions $\boldsymbol{T}_{EA}$. Thus we learn $\varepsilon_\theta$ by making per-node predictions $\varepsilon^k \in \mathbb{R}^7$ and optimising the variational lower bound of the data likelihood which has been shown (Ho et al., 2020) to be equivalent to minimising $MSE(\varepsilon^k - \varepsilon_\theta(\mathcal{G}^k))$. As our parameterised model, we use a heterogeneous graph transformer, which updates features of each node in the graph, $\mathcal{F}_i$, as (Shi et al., 2020):

$$\mathcal{F}_i' = \boldsymbol{W}_1\mathcal{F}_i + \sum_{j \in \mathcal{N}(i)} \text{att}_{i,j}\left(\boldsymbol{W}_2\mathcal{F}_j + \boldsymbol{W}_5 e_{ij}\right); \quad \text{att}_{i,j} = softmax\left(\frac{(\boldsymbol{W}_3\mathcal{F}_i)^T(\boldsymbol{W}_4\mathcal{F}_j + \boldsymbol{W}_5 e_{ij})}{\sqrt{d}}\right) \tag{3}$$

Here, $\boldsymbol{W}_i$ represent learnable weights. Equipping our model with such a structured attention mechanism allows for selective and informative information aggregation which is propagated through the graph in a meaningful way, while ensuring that memory and computational complexity scales linearly with increasing context length (both $N$ and $L$). More details can be found in Appendix C.

**Deployment.** During test time, we create the graph representation using actions sampled from the normal distribution, together with the current observation and the demonstrations as the context. We then make predictions describing how gripper nodes should be adjusted, and update the positions of these nodes by taking a denoising step according to the DDIM (Song et al., 2020):

$$p_g^{k-1} = \sqrt{\alpha_{k-1}}\hat{p}_g^0 + \sqrt{\frac{1 - \alpha_{k-1}}{1 - \alpha_k}}\left(p_g^k - \sqrt{\alpha_k}\hat{p}_g^0\right). \tag{4}$$

Here, $\hat{p}_g^0 = p_g^k + \Delta p_t + \Delta p_r$. This leaves us with two sets of points $p_g^{k-1}$ and $p_g^k$, that implicitly represent gripper poses at denoising time steps $k - 1$ and $k$. As we know the ground truth correspondences between them, we can extract an $\mathbb{SE}(3)$ transformation that would align them using a Singular Value Decomposition (SVD) (Arun et al., 1987) as:

$$\boldsymbol{T}_{k-1,k} = \underset{\boldsymbol{T}_{k-1,k} \in \mathbb{SE}(3)}{\arg\min} ||p^{k-1} - \boldsymbol{T}_{k-1,k} \times p^k||^2 \tag{5}$$

Finally, the $\boldsymbol{a}^{k-1}$ is calculated by applying calculated transformation $T_{k-1,k}$ to $\boldsymbol{a}^k$. Note that for gripper opening and closing actions utilising Equation 4 directly is sufficient. This process is repeated $K$ times until the graph that is in distribution is generated and, as a byproduct, final $\boldsymbol{a}^0$ actions are extracted, allowing us to sample from the initially described distribution $p(\boldsymbol{a} \mid o_t, C)$.

### 3.4 AN INFINITE POOL OF DATA

Now that we can learn the conditional distribution of actions, we need to answer the question of where a sufficiently large and diverse dataset will come from, to ensure that the learned model can be used for a wide range of real-world tasks. With In-Context Learning, the model does not need to encode task-specific policies into its weights. Thus it is possible to simulate 'arbitrary but consistent' trajectories as training data. Here, consistent means that while the trajectories differ, they 'perform' the same type of *pseudo-task* at a semantic level. We call such trajectories *pseudo-demonstrations*.

**Data Generation.** Firstly, to ensure generalisation across object geometries, we populate a simulated environment using a diverse range of objects from the ShapeNet dataset (Chang et al., 2015). We then create pseudo-tasks by randomly sampling object-centric waypoints near or on the objects, that the robot needs to reach in sequence. Finally, by virtually moving the robot gripper between them and occasionally mimicking rigid grasps by attaching objects to the gripper, we create pseudo-demonstrations – trajectories that resemble various manipulation tasks. Furthermore, randomising the poses of the objects and the gripper, allows us to create many pseudo-demonstrations performing the same pseudo-task, resulting in the data that we use to train our In-Context model.

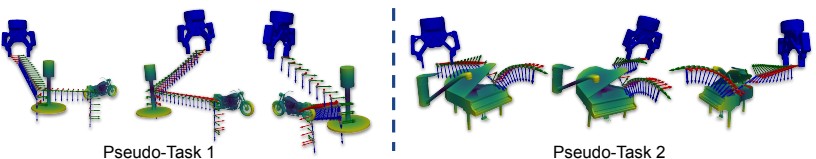

Pseudo-Task 1        Pseudo-Task 2

Figure 4: Examples of the simulated trajectories - 3 pseudo-demonstrations for 2 pseudo-tasks.

In practice, to facilitate more efficient learning of common skills, we bias sampling towards waypoints resembling tasks like grasping or pick-and-place. Note that as the environment dynamics and task specifications, such as feasible grasps, are defined as context at inference, we do not need to ensure that these trajectories are dynamically or even kinematically feasible. In theory, with enough randomisation, the convex hull of the generated trajectories would encapsulate all the possible test-time tasks. More information about the data generation process can be found in Appendix D.

**Data Usage.** During training, we sample $N$ pseudo-demonstrations for a given pseudo-task, using $N-1$ to define the context while the model learns to predict actions for the $N$th. Although pseudo-demonstrations are the primary training data, our approach can integrate additional data sources in the same format, allowing the model to adapt to specific settings or handle noisier observations.

## 4 EXPERIMENTS

We conducted experiments in two distinct settings: 1) simulation with RLBench (James et al., 2020) and ground truth segmentations, and 2) real-world everyday tasks. Our experiments study performance relative to baselines, to understand the effect of different design choices, to reveal the scaling trends, and to showcase applicability in cross-embodiment and modality transfer. Videos are available on our anonymous webpage at https://www.robot-learning.uk/instant-policy.

**Experimental Setup.** In all our experiments, unless explicitly stated otherwise, we use a single model to perform various manipulation tasks by providing N=2 demos, expressed as L=10 waypoints as context and predict T=8 future actions. We train this model for 2.5M optimisation steps using pseudo-demonstrations that are continuously generated in parallel, which is roughly equivalent to using 700K unique trajectories. During training, we randomise the number of demos in context between 1 and 5. When we discuss integrating additional training data beyond pseudo-demonstrations, we refer to models fine-tuned for an additional 100K optimisation steps using a 50/50 mix of pseudo-demonstrations and new data. For more information, please refer to Appendix E.

**Baselines.** We compare Instant Policy to 3 baselines which also enable In-Context Imitation Learning, namely: BC-Z (Jang et al., 2022), Vid2Robot (Jain et al., 2024), and a GPT2-style model (Radford et al., 2019). BC-Z combines latent embedding of the demonstrations with the current observation and uses an MLP-based model to predict robot actions, Vid2Robot utilises a Perceiver Resampler (Jaegle et al., 2021) and cross-attention to integrate information from the context and current

observation, and GPT2 uses causal self-attention to predict the next tokens in the sequence, which in our case are robot actions. For a fair comparison, we implemented all baselines by adapting them to work with point cloud observation using the same pre-trained encoder, and all have roughly the same number of trainable parameters. Additionally, all components that rely on language-annotated data, such as auxiliary losses, were removed because our generated pseudo-demonstrations do not have such information. To highlight this, we add an asterisk to these methods when discussing results.

## 4.1 SIMULATED EXPERIMENTS

The aim of our first set of experiments is two-fold: 1) to evaluate the effectiveness of Instant Policy in performing various manipulation tasks by comparing it to state-of-the-art baselines, and 2) to investigate the role the training data plays in generalising to unseen scenarios. We use a standard RLBench setup using the Franka Emika Panda and test Instant Policy (IP) and the baselines on 24 tasks, 100 rollouts each, randomising the poses of the objects in the environment each time. Additionally, we test models trained using only pseudo-demonstrations (PD only) and a combination of pseudo-demonstrations and 20 demonstrations for each of 12 out of the 24 RLBench tasks (PD++).

**Results & Discussion.** The first notable observation from the results, presented in Table 1, is that all methods achieve non-zero success rates when only pseudo-demonstrations are used and can perform well on at least the simpler tasks. This indicates that these pseudo-demonstrations are a powerful source of limitless data for In-Context Imitation Learning. Our second observation is that incorporating additional demonstrations from the same domain can greatly boost the performance, helping with generalisation to unseen tasks and novel object poses. Our third observation is that Instant Policy achieves significantly higher success rates than the baselines, showing the importance of our graph representation and its ability to interpret the context. We further demonstrate this by visualising attention weights (Figure 5), which reveal the model's ability to understand the task's current stage and identify the relevant information in the context. We discuss failure cases in Appendix G.

| Tasks | Instant Policy | BC-Z* | Vid2Robot* | GPT2* | Tasks | Instant Policy | BC-Z* | Vid2Robot* | GPT2* |
|---|---|---|---|---|---|---|---|---|---|
| Open box | 0.94 / 0.99 | 0.22 / 0.98 | 0.30 / 0.97 | 0.25 / 0.95 | Slide buzzer | 0.35 / 0.94 | 0.04 / 0.26 | 0.05 / 0.19 | 0.00 / 0.00 |
| Close jar | 0.58 / 0.93 | 0.00 / 0.06 | 0.00 / 0.11 | 0.00 / 0.22 | Plate out | 0.81 / 0.97 | 0.26 / 0.55 | 0.11 / 0.52 | 0.31 / 0.40 |
| Toilet seat down | 0.85 / 0.93 | 0.40 / 0.88 | 0.54 / 0.85 | 0.38 / 0.83 | Close laptop | 0.91 / 0.95 | 0.64 / 0.65 | 0.45 / 0.57 | 0.53 / 0.72 |
| Close microwave | 1.00 / 1.00 | 0.55 / 0.60 | 0.72 / 0.87 | 0.76 / 1.00 | Close box | 0.77 / 0.99 | 0.81 / 1.00 | 0.89 / 0.88 | 0.42 / 0.41 |
| Phone on base | 0.98 / 1.00 | 0.51 / 0.50 | 0.48 / 0.51 | 0.28 / 0.55 | Open jar | 0.52 / 0.78 | 0.12 / 0.28 | 0.15 / 0.30 | 0.22 / 0.51 |
| Lift lid | 1.00 / 1.00 | 0.82 / 0.82 | 0.90 / 0.91 | 0.88 / 0.94 | Toilet seat up | 0.94 / 1.00 | 0.62 / 0.63 | 0.58 / 0.64 | 0.31 / 0.34 |
| Take umbrella out | 0.88 / 0.91 | 0.42 / 0.64 | 0.90 / 0.90 | 0.75 / 0.89 | Meat off grill | 0.77 / 0.9 | 0.75 / 0.64 | 0.76 / 0.33 | 0.80 / 0.30 |
| Slide block | 0.75 / 1.00 | 0.10 / 0.14 | 0.12 / 0.16 | 0.08 / 0.16 | Open microwave | 0.23 / 0.56 | 0.00 / 0.13 | 0.00 / 0.02 | 0.00 / 0.00 |
| Push button | 0.60 / 1.00 | 0.75 / 0.81 | 0.85 / 0.88 | 0.80 / 0.91 | Paper roll off | 0.70 / 0.95 | 0.32 / 0.53 | 0.29 / 0.55 | 0.26 / 0.48 |
| Basketball in hoop | 0.66 / 0.97 | 0.02 / 0.09 | 0.00 / 0.06 | 0.03 / 0.07 | Put rubish in bin | 0.97 / 0.99 | 0.11 / 0.11 | 0.12 / 0.14 | 0.18 / 0.17 |
| Meat on grill | 0.78 / 1.00 | 0.64 / 0.88 | 0.51 / 0.77 | 0.53 / 0.81 | Put umbrella | 0.31 / 0.37 | 0.35 / 0.34 | 0.41 / 0.28 | 0.28 / 0.39 |
| Flip switch | 0.40 / 0.94 | 0.15 / 0.63 | 0.05 / 0.16 | 0.11 / 0.42 | Lamp on | 0.42 / 0.41 | 0.00 / 0.00 | 0.00 / 0.00 | 0.00 / 0.00 |
| Average (PD++) Seen | **0.97** | 0.59 | 0.60 | 0.65 | Average (PD++) Unseen | **0.82** | 0.43 | 0.37 | 0.31 |
| Average (PD only) All | **0.71** | 0.36 | 0.38 | 0.34 | | | | | |

Table 1: Success rates for Instant Policy and baselines on 24 tasks. 100 rollouts for each (trained using only pseudo-demonstrations / with additional demos from the 12 RLBench tasks on the left).

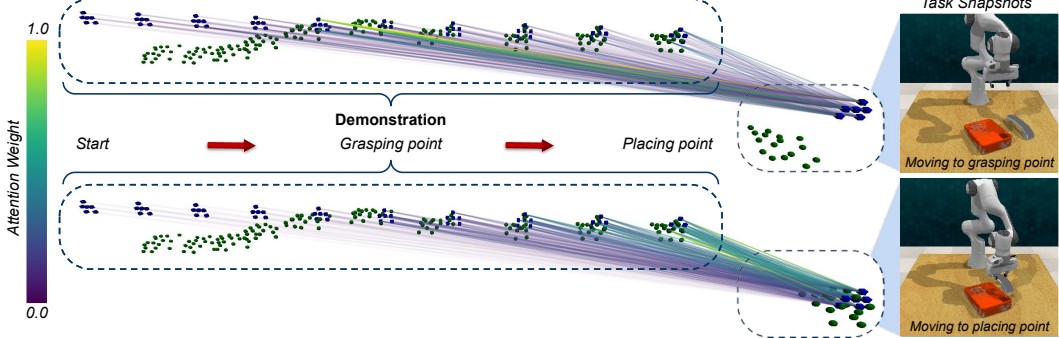

Figure 5: Attention weights visualised on sub-graph edges at two different timesteps in the phone-on-base task, showing the model's ability to track task progress and aggregate relevant information.

## 4.2 INSTANT POLICY DESIGN CHOICES & SCALING TRENDS

Our next set of experiments investigates the impact of various hyperparameters on the performance of our method, focusing on design choices requiring model re-training, inference parameters that

alter model behaviour at test time, and scaling trends as model capacity and training time increase. For the design choices and inference parameters, we calculate the average change in success rate on 24 unseen RLBench tasks, with respect to the base model used in the previous set of experiments, while for the scaling trends, we report validation loss on a hold-out set of pseudo-demonstrations to see how well it can capture the underlying data distribution.

| Design Choices | | | | | | Inference Parameters | | | | | |
|---|---|---|---|---|---|---|---|---|---|---|---|
| Action Mode | $\Delta\%$ | Diffusion Mode | $\Delta\%$ | Prediction Horizon (T) | $\Delta\%$ | # Diffusion Steps (K) | $\Delta\%$ | Demo Length (L) | $\Delta\%$ | # Demos (N) | $\Delta\%$ |
| $\Delta p$ | -15 | Flow | 0 | 1 | -52 | 1 | -16 | 1 | -71 | 1 | -12 |
| $(\Delta p_t, \Delta p_r)$ | 0 | Sample | -6 | 4 | -13 | 2 | -2 | 5 | -26 | 2 | 0 |
| $(\Delta t, \Delta q)$ | -37 | Noise | -7 | 8 | 0 | 4 | 0 | 10 | 0 | 3 | 2 |
| $(\Delta t, \Delta\theta)$ | -21 | No Diff | -29 | 16 | -4 | 8 | 0 | 15 | 1 | 4 | -1 |

Table 2: Performance change of ablation variants when compared to the base model.

**Design Choices.** We now examine the following: action mode, diffusion mode, and the prediction horizon. For action modes, we compare our proposed parameterisation, which decouples translation and rotation, against an approach without such decoupling, and more conventional approaches like combining translation with quaternion or angle-axis representations. For diffusion mode, we evaluate predicting flow versus added noise, direct sample, and omitting diffusion, regressing the actions directly. Lastly, we assess the impact of predicting different numbers of actions. The results, shown in Table 2 (left), show that these choices greatly influence performance. Decoupling translation and rotation in Cartesian space allows for precise low-level action learning. The diffusion process is vital for capturing complex action distributions, with predicting flow showing the best results. Finally, predicting multiple actions is helpful, but this also increases computational complexity. For a detailed discussion of other design choices, including unsuccessful ones, please refer to Appendix H.

**Inference Parameters.** Using a diffusion with a flexible representation that handles arbitrary context lengths allows us to adjust model performance at inference. We investigate the impact of the number of diffusion steps, the demonstration length, and the number of demonstrations in the context, as shown in Table 2 (right). Results show that even with just two denoising steps, good performance can be achieved. Demonstration length is critical; it must be dense enough to convey how the task should be solved, as this information is not encoded in the model weights. This is evident when only the final goal is provided (demonstration length = 1), leading to poor performance. However, extending it beyond a certain point shows minimal improvement, as the RLBench tasks can often be described by just a few waypoints. For more complex tasks, dense demonstrations would be crucial. Finally, performance improves with multiple demonstrations, though using more than two seems to be unnecessary. This is because two demonstrations are sufficient to disambiguate the task when generalising only over object poses. However, as we will show in other experiments, this does not hold when the test objects differ in geometry from those in the demonstrations.

**Scaling Trends.** The ability to continuously generate training data in simulation allows our model's performance to be limited only by available compute (training time) and model capacity (number of trainable parameters). To assess how these factors influence our approach, we trained three model variants with different numbers of parameters and evaluated them after varying numbers of optimisation steps (Figure 6). The results show that the model's ability to capture the data distribution (as reflected by decreasing validation loss) scales well with both training time and model complexity. This offers some promise that scaling compute alone could enable the development of high-performing models for robot manipu-

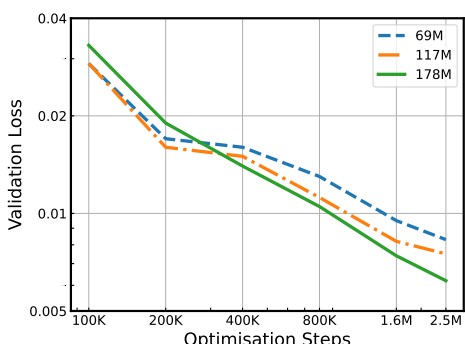

Figure 6: Validation loss curves for three different model sizes.

lation. Qualitatively, we observed a similar performance trend on unseen RLBench tasks. With increased training, we see an increase in performance. However, it plateaus eventually. Similarly, by increasing the model's capacity from 69M to 117M, the success rate reached before plateauing

increases significantly. However, further increasing the number of trainable parameters to 178M results in only minor, insignificant improvements in performance. This suggests the need for more diverse and representative data. Such data could come from available robotics datasets or the generation of pseudo-demonstrations that more closely resemble real tasks.

## 4.3 REAL-WORLD EXPERIMENTS

In real-world experiments, we evaluate our method's ability to learn everyday tasks and generalise to novel objects, unseen in both the training data and the context. We use a Sawyer robot with a Robotiq 2F-85 gripper and two external RealSense D415 depth cameras. We obtain segmentation by seeding the XMem++ (Bekuzarov et al., 2023) object tracker with initial results from SAM (Kirillov et al., 2023), and we provide demonstrations using kinesthetic teaching. To help the model handle imperfect segmentations and noisy point clouds more effectively, we further co-fine-tuned the model used in our previous experiments using 5 demos from 5 tasks not included in the evaluation.

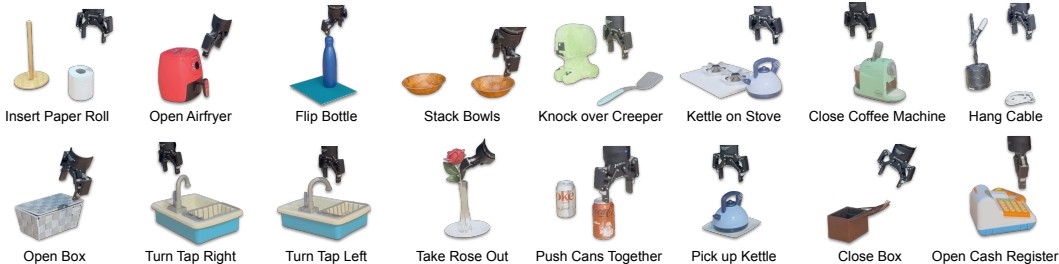

Figure 7: The 16 tasks used in our real-world evaluation.

**Real-World Tasks.** To evaluate our model's ability to tackle various tasks in the real world, we tested it and the baselines on 16 everyday tasks (Figure 7). We evaluated all methods using 10 rollouts, randomising the poses of the objects in the environment each time. From the results (Table 3), we can see that Instant Policy is able to complete various everyday tasks from just a couple of demonstrations with a high success rate, outperforming the baselines by a large margins.

| | Insert Paper Roll | Open Airfryer | Flip Bottle | Stack Bowls | Knock over Creeper | Kettle on Stove | Close Coffee Machine | Hang Cable | |
|---|---|---|---|---|---|---|---|---|---|
| Instant Policy | 9 / 10 | 9 / 10 | 7 / 10 | 10 / 10 | 8 / 10 | 10 / 10 | 10 / 10 | 7 / 10 | |
| BC-Z* | 1 / 10 | 5 / 10 | 0 / 10 | 2 / 10 | 5 / 10 | 1 / 10 | 1 / 10 | 0 / 10 | |
| Vid2Robot* | 3 / 10 | 6 / 10 | 0 / 10 | 1 / 10 | 7 / 10 | 3 / 10 | 4 / 10 | 1 / 10 | |
| GPT2* | 1 / 10 | 6 / 10 | 0 / 10 | 4 / 10 | 5 / 10 | 5 / 10 | 5 / 10 | 1 / 10 | |
| | Open Box | Turn Tap Right | Turn Tap Left | Take Rose Out | Push Cans Together | Pick up Kettle | Close Box | Open Register | Average, % |
| Instant Policy | 8 / 10 | 10 / 10 | 10 / 10 | 9 / 10 | 5 / 10 | 10 / 10 | 10 / 10 | 10 / 10 | **88.75** |
| BC-Z* | 8 / 10 | 2 / 10 | 3 / 10 | 0 / 10 | 2 / 10 | 10 / 10 | 7 / 10 | 8 / 10 | 34.38 |
| Vid2Robot* | 9 / 10 | 4 / 10 | 3 / 10 | 0 / 10 | 1 / 10 | 10 / 10 | 7 / 10 | 6 / 10 | 40.63 |
| GPT2* | 0 / 10 | 5 / 10 | 5 / 10 | 0 / 10 | 0 / 10 | 10 / 10 | 5 / 10 | 7 / 10 | 36.88 |

Table 3: Real-world success rates for Instant Policy and the baselines, with 10 rollouts each.

**Generalisation to Novel Objects.** While all of our previous experiments focused on evaluating our method's performance on the same objects used in the demonstrations, here we aim to test its ability to generalise to novel object geometries at test time. We do so by providing demonstrations (i.e., defining the context) with different sets of objects from the same semantic category, and testing on a different object from that same category. For the evaluation, we use four different tasks (Figure 8), each with six sets of objects (four for the demonstrations/context and two for evaluation). We evaluate our method with a different number

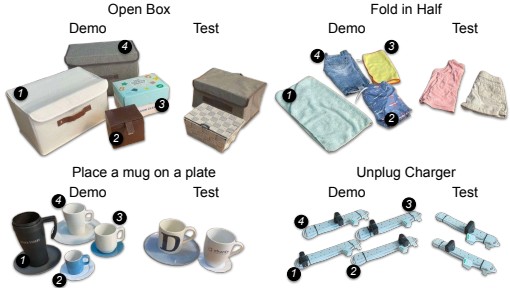

Figure 8: Objects used in the generalisation experiment (numbers indicate their usage stage).

of demonstrations in the context, randomising the poses of the test objects during each roll-out (5 rollouts for each unseen object set). The results, presented in Table 4, show that, with an increasing number of demonstrations across different objects, the performance on completely novel object geometries increases. This indicates that Instant Policy is capable of selectively aggregating and interpolating the information present in the context to disambiguate the task and the parts of the objects that are relevant to it. It is important to note that this is an emergent behaviour, as we never trained our model with objects from different geometries across the context, and is enabled by the graph representation and structured cross-attention mechanism.

### 4.3.1 Downstream Applications

**Cross-embodiment transfer.** Since our model uses segmented point clouds and defines the robot state by the end-effector pose and gripper state, different embodiments can be used to define the context and roll out the policy, provided the mapping between them is known. We demonstrate this by using human-hand demonstrations with a handcrafted mapping to the gripper state, allowing us to transfer the policy directly to the robot. In qualitative experiments, we achieve similar success rates on simple tasks, like pick-and-place, compared to kinesthetic teaching. However, for more complex tasks, this approach is limited by the handcrafted mapping. See our webpage for video examples and refer to Appendix I for more information about this mapping.

| N | Open Box | Fold in Half | Mug on Plate | Unplug Charger | Average, % |
|---|---|---|---|---|---|
| 1 | 2 / 10 | 7 / 10 | 7 / 10 | 0 / 10 | 40 |
| 2 | 5 / 10 | 8 / 10 | 8 / 10 | 0 / 10 | 52.5 |
| 3 | 10 / 10 | 10 / 10 | 9 / 10 | 5 / 10 | 85 |
| 4 | 10 / 10 | 10 / 10 | 9 / 10 | 7 / 10 | 90 |

Table 4: Success rates of Instant Policy with a different number of demonstrations (N), enabling generalisation to novel object geometries.

**Modality change.** While obtaining a policy immediately after demonstrations is a powerful and efficient tool, it still requires human effort in providing those demonstrations. We can circumvent this by exploiting the bottleneck of our trained model, which holds the information about the context and the current observation needed to predict actions. This information is aggregated in the gripper nodes of the current observations. If we approximate this bottleneck representation using different modalities, such as language, we can bypass using demonstrations as context altogether. This can be achieved with a smaller, language-annotated dataset and a contrastive objective. Using language-annotated trajectories from RLBench and rollout data from previous experiments, we qualitatively demonstrate zero-shot task completion based solely on language commands. For more details, see Appendix J, and for videos, visit our webpage at https://www.robot-learning.uk/instant-policy.

## 5 Discussion

**Limitations.** While Instant Policy demonstrates strong performance, it has several limitations. First, like many similar approaches, we assume the availability of segmented point clouds for sufficient observability. Second, point cloud observations lack colour or other semantically rich information. Third, our method focuses on relatively short-horizon tasks where the Markovian assumption holds. Fourth, Instant Policy is sensitive to the quality and downsampling of demonstrations at inference. Fifth, it does not address collision avoidance or provide end-to-end control of the full configuration space of the robot arm. Finally, it lacks the precision needed for tasks with extremely low tolerances or rich contact dynamics. However, we believe many of these limitations can be addressed primarily through improvements in generation of the pseudo-demonstrations, such as accounting for collisions, incorporating long-horizon tasks, and by improving the graph representation with additional features from vision models, force information, or past observations.

**Conclusions.** In this work, we introduced Instant Policy, a novel framework for In-Context Imitation Learning that enables immediate robotic skill acquisition following one or two test-time demonstrations. This is a compelling alternative paradigm to today's widespread behavioural cloning methods, which require hundreds or thousands of demonstrations. We showed that our novel graph structure enables data from demonstrations, current observations, and future actions, to be propagated effectively via a novel graph diffusion process. Importantly, Instant Policy can be trained with only pseudo-demonstrations generated in simulation, providing a virtually unlimited data source that is constrained only by available computational resources. Experiments showed strong performance relative to baselines, the ability to learn everyday real-world manipulation tasks, generalisation to novel object geometries, and encouraging potential for further downstream applications.

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

APPENDIX

## A   GEOMETRY ENCODER

Here, we describe the local geometry encoder used to represent observations of the environment as a set of nodes. Formally, the local encoder encodes the dense point cloud into a set of feature vectors together with their associated positions as: $\{\mathcal{F}^i, p^i\}_{i=1}^M = \phi(P)$. Here, each feature $\mathcal{F}^i$ describes the local geometry around the point $p^i$. We ensure this by pre-training an occupancy network (Mescheder et al., 2019), that consists of an encoder $\phi_e$, which embeds local point clouds, and a decoder $\psi_e$ which given this embedding and a query point is tasked to determine whether the query lays on the surface of the object: $\psi_e(\phi_e(P), q) \to [0, 1]$. The high-level structure of our occupancy network can be seen in Figure 9. Note that each local embedding is used to reconstruct only a part of the object, reducing the complexity of the problem and allowing it to generalise more easily.

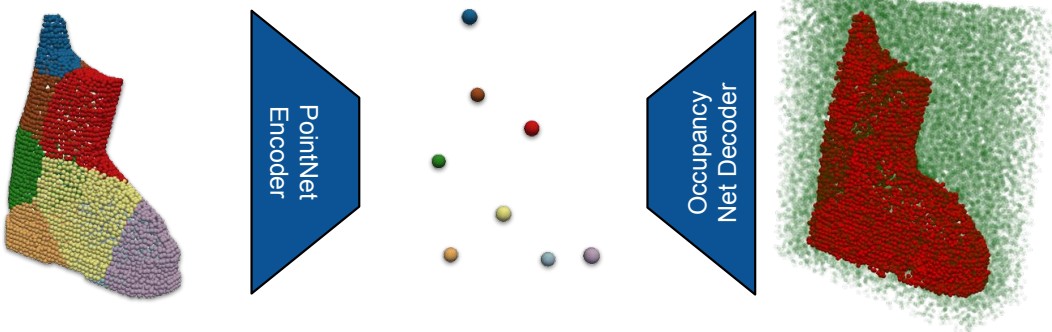

Figure 9: High-level structure of the occupancy network.

We parameterise $\phi_e$ as a network composed of 2 Set Abstraction layers (Qi et al., 2017) enhanced with Nerf-like sine/cosine embeddings (Mildenhall et al., 2021). It samples $M$ centroids from the dense point cloud and embeds the local geometries around them into feature vectors of size 512. Instead of expressing positions of points relative to the sampled centroids $p^i$ as $p_j - p^i \in \mathbb{R}^3$, we express them as $(sin(2^0\pi(p_j - p_i)), cos(2^0\pi(p_j - p_i)), ..., sin(2^9\pi(p_j - p_i)), cos(2^9\pi(p_j - p_i)))$, enabling the model to capture high-frequency changes in the position of the dense points and capture the local geometry more precisely. We parametrise $\psi_e$ as an eight-layer MLP with residual connections, that uses the same Nerf-like embeddings to represent the position of the query point. We use objects from a diverse ShapeNet (Chang et al., 2015) dataset to generate the training data needed to train the occupancy network. For training Instant Policy, we do not use the decoder and keep the encoder frozen.

## B   TRAINING

Training our diffusion model involves a forward and backward Markov chain diffusion process, which is outlined in Equations 1 and 2. Intuitively, we add noise to the ground truth robot actions and learn how to remove this noise in the graph space (see Figure 10).

In practice, training includes 4 main steps: 1) noise is added to the ground truth actions, 2) noisy actions are used to construct our graph representation, 3) the network predicts how nodes representing robot actions need to be adjusted to effectively remove the added noise, and 4) the prediction and ground truth labels are used to calculate the loss function, and weights of the network are updated accordingly.

To add noise to the action expressed as $(\boldsymbol{T}_{EA} \in \mathbb{SE}(3), a_g \in \mathbb{R}$, we first project $\boldsymbol{T}_{EA}$ to $se(3)$ using a Logmap, normalise the resulting vectors, add the noise as described by Ho et al. (2020), unnormalise the result and extract the noisy end-effector transformation $\boldsymbol{T}_{EA}^k$ using Expmap. Such a process can be understood as adding noise to a $\mathbb{SE}(3)$ transformation in its tangential space. We

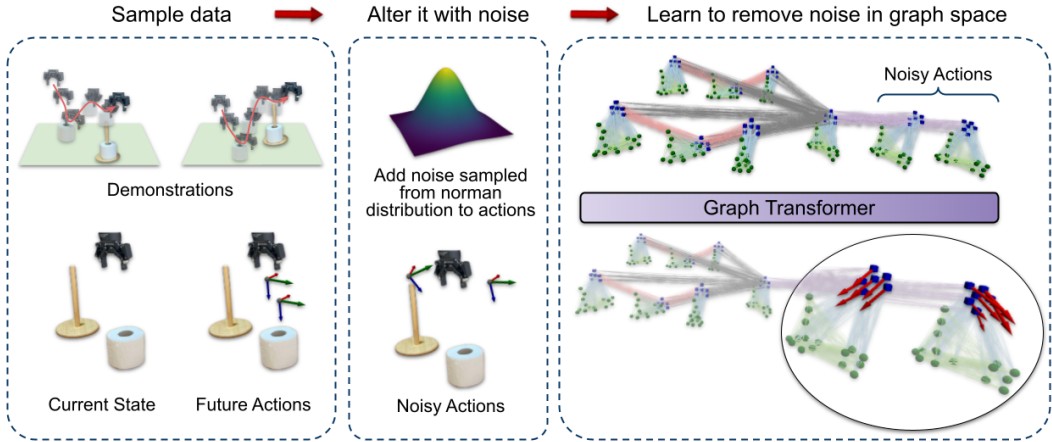

Figure 10: High-level overview of the training process. (Left) A data point is sampled from the dataset. (Middle) Noise is added to the ground truth actions. (Right) Using demonstrations, current observation and noisy actions, a graph representation is constructed, which is used to predict, how to remove the added noise in the graph space.

can do this because around actions (end-effector displacements) are sufficiently small. For bigger displacements, unnormalised noise should be projected onto the $\mathbb{SE}(3)$ manifold directly, as done by Vosylius & Johns (2023a) and Urain et al. (2023). Adding noise to real-valued gripper actions can be done directly using the process described by Ho et al. (2020).

## C  NETWORK ARCHITECTURE

Here we describe the neural network used to learn the denoising process on graphs, enabling us to generate graphs $\mathcal{G}$ and implicitly model the conditional action probability. Our parametrised neural network takes the constructed graph representation as input and predicts the gradient field for each gripper node representing the actions: $\varepsilon_\theta(\mathcal{G}^k)$. These predictions are then used in the diffusion process allowing to iteratively update the graph and ultimately extract desired low-level robot actions. In practice, for computational efficiency and more controlled information propagation, we are using three separate networks $\sigma$, $\phi$ and $\psi$, updating relevant parts of the graph in sequence as:

$$\varepsilon_\theta(\mathcal{G}^k) = \psi(\mathcal{G}(\sigma(\mathcal{G}_l^a), \phi(\mathcal{G}_c(\sigma(\mathcal{G}_l^t), \{\sigma(\mathcal{G}_l^{1:L})\}_1^N)))) \tag{6}$$

Here, $\sigma$ operates on local subgraphs $\mathcal{G}_l$ and propagates initial information about the point cloud observation to the gripper nodes, $\varphi$ additionally propagates information through the demonstrated trajectories and allows all the relevant information from the context to be gathered at the gripper nodes of the current subgraph. Finally, $\psi$ propagates information to nodes in the graph representing the actions. Using such a structured and controlled propagation of information through the graph, together with the learnable attention mechanism described in Equation 3, allows the model to continuously aggregate relevant information from the context and make accurate predictions about the actions. Additionally, it also results in a clear and meaningful bottleneck in the network with all the relevant information from the context aggregated in a specific set of nodes ($\phi(\mathcal{G}_c(\sigma(\mathcal{G}_l^t), \{\sigma(\mathcal{G}_l^{1:L})\}_1^N))$). This bottleneck representation could be used for retrieval or as shown in our experiments, to switch modalities, for example to language, via a smaller annotated dataset and a contrastive objective.

Each of the three separate networks is a heterogeneous graph transformer (Equation 3) with 2 layers and a hidden dimension of size 1024 (16 heads, each with 64 dimensions). As we are using heterogeneous graphs, each node and edge type are processed with separate learnable weights and aggregated via summation to produce all node-wise embeddings. This can be understood as a set of cross-attention mechanisms, each responsible for processing different parts of the graph representation. We use layer normalisation layers (Ba, 2016) between every attention layer and add additional

residual connections to ensure good propagation of gradients throughout the network. Finally, features of the nodes representing robot actions are processed with a 2-layer MLP equipped with GeLU activations (Hendrycks & Gimpel, 2016) to produce the per-node denoising directions.

## D    DATA GENERATION

Our data generation process, firstly, includes populating a scene with objects with which the robot will interact. We do so by sampling two objects from the ShapeNet dataset and placing them randomly on a plane. Next, we define a pseudo-task by sampling a sequence of waypoints on or near those objects. The number of these waypoints is also randomly selected to be between 2 and 6, inherently modelling various manipulation tasks. We assign one or more waypoints to change the gripper state, mimicking the rigid robotic grasp and release. We then sample a starting pose for the gripper, where we initialise a mesh of a Robotiq 2F-85 gripper. By moving the gripper between the aforementioned waypoints and attaching or detaching the closest object to it when the gripper state changes, we create a pseudo-demonstration. To further increase the diversity in pseudo-tasks, we use different interpolation strategies between the waypoints (e.g. linear, cubic or interpolating while staying on a spherical manifold). We record gripper poses and segmented point cloud observations using PyRender (Matl, 2019) and three simulated depth cameras. We ensure that the spacing between the subsequent spaces is constant and uniform (1cm and 3 degrees, same as used for the normalisation of actions). Moving objects to different poses, choosing a different starting gripper pose and repeating the process results in several pseudo-demonstrations for the same pseudo-task, which we use to train our In-Context model. As mentioned in Section 3.4, we do not need to ensure that these generated trajectories are dynamically or even kinematically feasible, as the environment dynamics and task specifications, such as feasible grasp, are defined as context at inference.

**Bias Sampling.** To facilitate more efficient learning of common skills, we bias the sampling to favour waypoints resembling common tasks such as grasping or pick-and-place. This does not require creating dynamically feasible trajectories but rather involves designing sampling strategies for waypoints that loosely approximate these tasks. For instance, by selecting a specific part of an object, moving the simulated gripper to that location, and closing the gripper, we can simulate a grasping task, even if the grasp itself is not physically feasible. We design such sampling strategies for common tasks, such as grasping, pick-and-place, opening or closing. Pseudo-demonstrations are generated using these strategies for half of the samples, while the rest use completely random waypoints.

**Data Augmentation.** To facilitate the emergence of recovery behaviour of the learnt policies, we further augment the generated trajectories. Firstly, for $30\%$ of the trajectories, we add local disturbances associated with actions that would bring the robot back to the reference trajectory, similarly to how it is done by Zhou et al. (2023). Secondly, for $10\%$ of the data points, we purposely change the gripper's open-close state. This, we found to be crucial, as, without it, the policy would never try to re-grasp an object after initially closing the gripper.

## E    IMPLEMENTATION DETAILS

Here we discuss the implementation details of the Instant Policy, which we found to be important in making the method perform well.

**Demo Processing.** Although our network can handle an arbitrary number of demonstrations of any length, we downsample the demo trajectories to a fixed length ($L = 10$ in our experiments). First, we record demonstrations at a rate of 25Hz and 10Hz in simulation and the real world, respectively. The lower rate in the real world is caused by the simultaneous segmentation of objects of interest by Xmem++ (Bekuzarov et al., 2023). We then include the start and end of the trajectories and all the waypoints where the open-close state of the gripper changed. We then include the waypoints in the trajectory, where the gripper sufficiently slowed down, indicating important trajectory stages (similar to Shridhar et al. (2023)). Finally, if the current number of the trajectory waypoint is less than $L$, we add intermediate waypoints between the already extracted ones.

**Data Augmentation.** To achieve robust policies, we found that it is crucial to randomise current observations and subsequent actions during training. The network can easily overfit to binary gripper

states (if it is open, just keep it open, and if it is closed, just keep it closed). To tackle this, we, with a 10% probability, flip the current gripper state used as an input to the model. This greatly increased the robustness of the resulting policies. Additionally, during the pseudo-demonstration generation process, we added local perturbations to the pose of the gripper (adjusting point cloud observations and actions accordingly), further increasing robustness and enabling recovery behaviour.

**Normalisation.** We normalise all the outputs of our mode to be $[-1, 1]$, a step that we found to be crucial. To this end, we manually define the maximum end-effector displacement between subsequent action predictions to be no more than $1cm$ in translation and $3\deg$ in rotation and clamp the noisy actions to be within this range. Thus the flow prediction is capped to be at most twice the size of this range. Knowing this, we normalise $\nabla \hat{p}_t$ and $\nabla \hat{p}_r$ to be between $-1$ and $1$ independently, enabling efficient network training. For the gripper opening-closing actions, this can be done easily as they are expressed as binary states $\{0, 1\}$. We do not normalise the position of the point cloud observations but rather rely on the sine/cosine embeddings, a strategy that we found to be sufficient.

**Point Cloud Representation.** We use segmented point cloud observations of objects of interest in the environment as our observations. These segmented point clouds do not include the points on the robot or other static objects such as the table or distractors. In practice, We downsample the point clouds to contain $2048$ points and express them in the end-effector frame as $\boldsymbol{T}_{EW} \times P$ to achieve stronger spatial generalisation capabilities. These point clouds are then processed with a geometry encoder, described in Section A, producing $M = 16$ nodes used to construct our devised graph representation.

**Action Denoising.** When updating $\boldsymbol{T}_{EA}$ during our denoising process, we use calculated $\boldsymbol{T}_{k,k-1}$ (as described in Section 3.3) and calculate the transformation representing end-effector actions during the denoising process as $\boldsymbol{T}_{EA}^{k-1} = \boldsymbol{T}_{k,k-1} \times \boldsymbol{T}_{EA}^{k}$. These actions are then used to construct a graph representation that is used in the next denoising step. In practice, because we express point cloud observations in the end-effector frame, we apply the inverse of these actions to the $M$ points representing the scene and construct local graphs of actions as $\mathcal{G}_l^a(\boldsymbol{T}_{EA}^{-1} \times P^t, \boldsymbol{T}_{WE}^t, a_g)$. As there are no absolute positions in the graph, this is equivalent to applying the actions to the gripper pose $\boldsymbol{T}_{WE}^t$, but it allows us to recompute the geometry embeddings of the point clouds at their new pose, better matching the ones from the demonstrations.

**Training.** We trained our model using AdamW (Loshchilov, 2017) optimiser with a $1e^{-5}$ learning rate for 2.5M optimisation steps (approx. 5 days on a single NVIDIA GeForce RTX 3080-ti) followed by a 50K steps learning rate cool-down period. For efficient training, we used float16 precision and compiled our models using torch compile capabilities (Paszke et al., 2019). Training data in the form of pseudo-demonstrations were continuously generated during training, replacing the older sample to ensure that the model did not see the same data point several times, preventing overfitting.

# F    SIMULATION EXPERIMENTAL SETUP

Here, we describe the **2** changes we made to a standard RLBench setup (James et al., 2020) when conducting our experiments. **1)** We generated all the demonstrations (for the context and for those used during training as described in Section 4) using only Cartesian Space planning - we disregarded all demonstrations that were generated using an RRT-based motion planner (Kuffner & LaValle, 2000). We did so to ensure that the demonstrations did not have arbitrary motions that would not be captured by our observations of segmented point clouds and end-effector poses. **2)** We restricted the orientations of the objects in the environment to be within $[-\pi/3, \pi/3]$. We did so to match the distribution of object poses to the one present in our generated pseudo-demonstrations. It also ensured that most tasks could be solved without complex motions requiring motion planners.

# G    FAILURE CASES

Here we discuss the observed failure modes of Instant Policy during our experiments. Given different setups and assumptions, we do so for each of our experiments independently. However, the discussed failure modes are shared across the experiments.

**Simulated Experiments.** During our simulated experiments using RLBench (James et al., 2020), we observed several common failure modes of Instant Policy. First of all, tasks such as Open Microwave or Put Umbrella into a Rack require extremely high precision in action predictions, otherwise, the inaccurate dynamics of the simulator will prevent the task from being completed. As such, sometimes the handle of the microwave would slip from the gripper, or the umbrella would fly off when in contact with the robot. Second, tasks such as Flipping a Switch or Pushing a Button terminate immediately after the task condition is met. As we predict actions of not doing anything at the end of the trajectory, this resulted in the policy stopping before the task is fully completed at a state virtually the same as the desired one. Moreover, our generated pseudo-demonstrations do not include any collision avoidance, which has proven to be a problem for tasks such as Turning the Lamp On, where the robot occasionally collides with the lamp by moving in a straight line towards the button. Finally, other failure modes usually included policy stalling at a certain point or oscillating between two configurations. We hypothesise that such behaviour is caused by conflicting information in the provided demonstrations and violating the Markovian assumption. In the future, this could be addressed by incorporating past observations into the graph representation.

**Real-World Tasks.** By far, the most common failure mode in our real-world experiments was the segmentation failure caused by several occlusions. Additionally, imperfect segmentation sometimes included parts of the robot or the table, causing the policy to perform irrelevant actions. This also sometimes degraded the quality of the demonstrations by including irrelevant points (and thus nodes in the constructed graph). Moreover, we observed that the overall quality of the demonstrations, in terms of smoothness and clearly directed motions, had a major impact on the performance of Instant Policy. If recorded demonstrations included inconsistent and arbitrary motions, information in the context was conflicting, resulting in the policy stalling or oscillating. Finally, other observed failure cases mainly involved policy not completing the task due to the lack of precision.

**Generalisation to Novel Geometries.** When evaluating Instant Policy using objects unseen neither during training nor demonstration at inference, policy sometimes just mimicked the motion observed during the demonstrations without achieving the desired outcome. With an increasing number of diversity demos in the context, such behaviour was minimised. However, some tasks (e.g. placing a mug on a plate) were completed mainly due to the high tolerance of the task rather than true generalisation capabilities.

**Cross-Embodiment Transfer.** The main failure cases during our cross-embodiment transfer experiments were caused by incorrect mapping of hand poses to end-effector poses and an insufficient field of view in our observations. This caused the robot to occasionally miss the precise grasping locations, closing the gripper at stages where it was not intended, and, in general, resulted in demonstrations of poorer quality.

**Modality Transfer.** Replacing demonstrations with language descriptions of the task yielded promising results in our qualitative experiments. However, the observed behaviour was sometimes mismatched with the object geometries in the environment. For instance, the policy would execute appropriate motions (e.g., pushing or closing) but at incorrect locations relative to the objects. This issue likely stems from object features containing only geometric information without any semantic context. Incorporating additional features from vision foundation models into the point cloud observations and expanding the language-annotated dataset could help address this limitation.

## H    THINGS THAT DID NOT WORK

Here we discuss various design choices we considered before settling for the approach, described in Section 3.

**Fully-Connected Graph.** Initially, we experimented with a fully connected graph, which effectively acts as a transformer with dense self-attention. While the attention mechanism should, in theory, learn the structure relevant to the task, this approach failed to produce good results, even for simple tasks.

**One Big Network.** Instead of using three separate networks in sequence (as described in Section C), we experimented with a single larger network, which led to a significant drop in performance. We hypothesize that this is because, early on, the nodes lack sufficient information to reason about ac-

tions, causing much of the computation to be wasted and potentially resulting in conflicting learning signals.

**More Gripper Nodes.** We express the robot state as a set of six nodes in the graph. In theory, we can use an arbitrary number ($> 3$) of such nodes, allowing more flexible aggregation of relevant information. We experimented with different numbers of such nodes and observed minimal changed in performance, while the computational requirements increased significantly.

**No Pre-trained Geometry Encoder.** During the training of Instant Policy, we keep the geometry encoder frozen. We experimented with training this model from scratch end-to-end, as well as fine-tuning it. Training from scratch did not work at all, while fine-tuning resulted in significantly worse performance. We also experimented with larger sizes of the encoder and saw no improvement, indicating that the geometry information was already well represented.

**Homogeneous Graph.** Instead of using a heterogenous graph transformer, which processes different types of nodes and edges using separate sets of learnable weights, we tried using a homogeneous variant with distinct embeddings added to the nodes and edges. This approach resulted in significantly worse performance, given the same number of trainable parameters. This indicates that by not sharing the same weights, different parts of the network can better focus on aggregating and interpreting relevant information, resulting in more efficient learning.

**Predicting Waypoints.** Initially, we tried predicting spare waypoints instead of low-level actions, e.g. velocities, that progress the execution of a task. We found, that because of these waypoints represent larger end-effector displacements, predicting them with high precision was challenging. Intuitively, this is the result of the increased action space that, when normalised, needs to be represented in an interval $[-1, 1]$.

**Larger Learning Rates.** For our experiments, we used a relatively small learning rate of $1e^{-5}$. To speed up the training process, we tried increasing it. However, with increased learning rate we found the training process to be unstable, resulting in large gradients and increasing training loss. We also tried using several different optimisers, using AdamW (Loshchilov, 2017) resulting in the best performance.

## I   CROSS-EMBODIMENT TRANSFER

As described in Section 4.3.1, our approach allows us to provide demonstrations using one embodiment (e.g. using human hands) and instantly deploy a policy on a robot, given a mapping between different embodiments is known. This is because our observations are composed of segmented point clouds that do not include points on the robot and its end-effector pose. Thus, by mapping the pose of a human hand to the robot's end-effector pose, we can effectively obtain the same observations. In our experiments, we achieve this mapping using a hand keypoint detector from Mediapipe (Lugaresi et al., 2019) and manually designing a mapping between these key points and the corresponding robot's end-effector pose. We model the position of the end-effector to be represented by the midway position between the index finger and the thumb and estimate the orientation using an additional point on the palm of the hand. In this way, we effectively overparametrise the $\mathbb{SE}(3)$ pose of the hand, modelled as a parallel gripper, using a set of positions. This allows us to complete simple tasks, such as grasping or pick-and-place. However, for more precise tasks, such a crude mapping can be insufficient. It could be addressed by using more elaborate mappings between human hands and robot grippers, for example, as done by Papagiannis et al. (2024).

## J   MODALITY TRANSFER

Using our graph representation together with network architecture, discussed in Section C, results in a clear information bottleneck with all the relevant information from the context aggregated in a specific set of nodes ($\phi(\mathcal{G}_c(\sigma(\mathcal{G}_l^t), \{\sigma(\mathcal{G}_l^{1:L})\}_1^N))$). Information present in the nodes of the graph representing the current information holds all the necessary information to compute precise robot action appropriate for the current situation, and a trained $\psi(.)$ has the capacity to do it. We exploit this bottleneck and learn to approximate it using the current observation and a language description of a task and utilise a frozen $\psi(.)$ to compute the desired robot actions in the same way as done when the context includes demonstrations. We learn this approximation using the local graph rep-

resentation of the current observation $\mathcal{G}_l^t$ and a language embedding of the task description $f_{lang}$, produced by Sentence-BERT (Reimers, 2019). We use a graph transformer architecture, similar to the one used to learn $\sigma$, and incorporate $f_{lang}$ as an additional type of node in the graph. We train this network using a language-annotated dataset comprising demonstrations from RLBench and rollouts from our experiments, along with a contrastive objective. At inference, we provide a language description of a task and, based on the current observation, compute the embeddings of the aforementioned bottleneck. We then use it to compute robot actions that are executed closed-looped, allowing for zero-shot generalisation to tasks described by language. Although showing promising performance using only a small language-annotated dataset, further improvements could be achieved by incorporating semantic information into the observation, using a variational learning framework and expanding the dataset size. We leave these investigations for future work.

