# OpenReview forum: "Instant Policy: In-Context Imitation Learning via Graph Diffusion"
_ICLR.cc/2025/Conference — ICLR 2025 Oral_

### Official Review · Reviewer_Wb4x · 2024-10-22

**Soundness:** 3
**Presentation:** 3
**Contribution:** 3
**Rating:** 8
**Confidence:** 3

**Summary:**

The paper presents a novel method for rapid robot policy learning. The proposed "Instant Policy" framework allows robots to instantly learn tasks from just one or two demonstrations by modeling in-context imitation learning (ICIL) as a graph-based diffusion process. The approach uses graph representations for demonstrations, observations, and actions, combined with a learned diffusion process, enabling efficient and scalable task learning. It can be trained using pseudo-demonstrations, offering a virtually infinite source of training data, and achieves high success rates on various tasks in both simulated and real-world environments.

**Strengths:**

1. The paper presents a substantial amount of experiments in both simulator and real-world, showcasing the effectiveness of the proposed method.
2. The proposed method demonstrates good performance in adapting to unseen tasks.
3. The idea of using pseudo-demonstrations to ensure that the learned network can be used for a wide range of real-world tasks is interesting.

**Weaknesses:**

1. I think there is a notation mistake in Line 163 $\{\mathcal{F}_g^i,p_g^i\}^6_{i=1}$.
2. Since the method consists of three kinds of graphs, the ablation study is missing.
3. The method section is hard to follow, it will be better with a clearer illustration figure.

**Questions:**

See weakness.

---

> ### Author Response · Authors · 2024-11-17
> **Response to Reviewer Wb4x**
>
> **Regarding the description of robot state representation and its notation.**
>
> Thank you for thoroughly checking our notations and raising your concerns. However, we believe that the notation is consistent and correct. We represent the gripper’s state as a collection of the positions of key points on the end-effector with their associated feature vectors. A set of positions is an over-parametrization of an SE(3) pose, while distinct feature vectors allow us to distinguish them while also holding information about the binary open-close gripper state. Thus a set of these points and feature vectors fully describe the state of the end-effector.
>
> **Regarding the graph representation and its ablations.**
>
> Although our graph representation consists of three separate parts describing different components of In-Context Imitation Learning (demonstrations, current observation and future actions), together they form a single graph that is used by our model. In Appendix G (Things that Did not Work), we describe other graph structures that failed to produce good results. Namely, a fully connected graph and a homogeneous graph. We did not include these results in the ablations section as the results were not very informative and left it as a qualitative discussion in the appendix.
>
> **Regarding improvements in describing our method.**
>
> Thank you for suggesting that the paper could be improved and made easier to understand. We have updated parts of the paper that could cause confusion (namely, the action representation, training procedure, use of the generated pseudo-demonstrations, and scaling trends) and added an additional figure to better convey the training procedure to the appendix. We also added new videos and visualisation on our anonymous webpage (https://sites.google.com/view/instant-policy/), that can help to understand our work better. If there are any particular parts of the paper that would benefit from further improvements, please let us know.

---

> > ### Comment · Reviewer_Wb4x · 2024-11-21
> >
> > Thanks for the response! I've looked at the other reviews/rebuttal responses; I think the explanation and improved clarity make the paper a solid one to accept.
> >
> > I'm not familiar enough with this area to call this work "groundbreaking" so I will maintain my original score, but I think this paper should be accepted.

---

> > > ### Author Response · Authors · 2024-11-25
> > > **Response to Reviewer Wb4x**
> > >
> > > Thank you for spending your time reviewing our paper, reading the responses and helping us improve our work! We are happy to hear that the new additions to the paper improved its quality and overall clarity. We are glad that you enjoyed our work and see the potential it holds.

---

### Official Review · Reviewer_ibtP · 2024-11-02

**Soundness:** 2
**Presentation:** 3
**Contribution:** 2
**Rating:** 8
**Confidence:** 4

**Summary:**

In the presented work, the authors introduce Instant Policy, a novel approach for in-context imitation learning (ICIL). The key contribution of this work is that demonstrations are represented as a graph, including the observations and actions trained from pseudo-demonstrations generated procedurally in simulation. Furthermore, given a single demonstration of the target task, the proposed method is able to generate a suitable control policy by re-framing the ICIL problem as a diffusion-based graph generation problem. The presented work is interesting for the perspective of not requiring data for the target manipulation tasks but is instead able to use an approach similar to learning from play.

**Strengths:**

- The graph-based diffusion process is interesting, particularly how the robot's motion is generated through the iterative diffusion process.
- The graph diffusion process in section 3.3 is well described, and generally, the paper is relatively easy to read.
- The results of the presented method are convincing, consistently outperforming the chosen baselines.
- The modality change in section 4.3.1 is very interesting and should be expanded on (the cross-embodiment transfer could be placed in the appendix instead)

**Weaknesses:**

- In section 3.2, it is unclear how actions are encoded. The graph seems to be encoding relative spatial relationships between key-points on task-relevant objects and the gripper, however, the "Action Representation" section is intuitively unclear. How are the position-based embeddings learned with respect to the actions? What exactly are the actions?
- Given that demonstrations are encoded as a sequence of relative spatial relationships, why are action nodes needed? Wouldn't it be possible to just "condition" the motion on the current observation with respect to the demonstrated graphs and optimize some form of minimal deformation of the graph? (similar to Vogt, David, et al. "A system for learning continuous human-robot interactions from human-human demonstrations." _2017 IEEE International Conference on Robotics and Automation (ICRA)_. IEEE, 2017.) (this might be a candidate for some discussion in the related work)
- The authors claim that, unlike BC, their method does not require much data; however, generating pseudo-demonstrations in simulation is also an expensive process. How does generating pseudo-demonstrations compare to alternative approaches, e.g., this one (Lynch, Corey, et al. "Learning latent plans from play." _Conference on robot learning_. PMLR, 2020.)? (this might be a candidate for some discussion in the related work)
- The generalization to novel objects within the same semantic group seems to stem from the model's ability to generalize due to the selection of only a few key points in the point cloud, making the "novelty" of the semantically similar objects questionable. How does the model handle novel objects from semantic groups that were never part of the training data, which would be a common occurrence in household tasks?

**Questions:**

- How are sub-goals formulated in the diffusion process? Is it just some horizon the diffusion model is able to generate "in one go"?
- How many pseudo-demonstrations are used to train the initial model across how many objects?
- Transferring the policy from simulation to the real robot seemed to have required only 25 examples (5 rollouts across 5 tasks). How was performance assessed, and would more samples result in significantly better results?

---

> ### Author Response · Authors · 2024-11-17
> **Response to Reviewer ibtP**
>
> **Regarding how actions are represented.**
>
> Thank you for pointing out how our paper could be improved and made easier to understand. In general, we represent a robot action as the relative transformation of the end-effector that would move the gripper from its current pose to a new desired pose and a binary open-close gripper action. These transformations are restricted to, at most, moving the gripper 1cm and rotating it by 3 degrees between subsequent timesteps, and, when time-scaled, can be interpreted as end-effector velocities.
>
> To incorporate such actions into our graph representation, we transform the nodes representing the gripper, moving them to the pose they would reach if the actions were executed, and set their features based on binary gripper actions. Edges between the nodes representing the current observation and those representing the actions are equipped with positional embeddings that encode the gripper's movement between timesteps. These embeddings are not learned but are expressed as a combination of sine and cosine functions, similar to the approach used in the original NeRF paper. This integration allows low-level robot actions to be represented in the graph, with the end-effector's motion captured through the gripper nodes' positions and binary gripper actions encoded in their features. We have updated our paper to convey this more clearly.
>
> At inference, our goal is to move the nodes representing robot actions to their desired position and adjust their features such that they correspond to correct binary gripper actions. This is done using a diffusion process, starting from randomly initialised actions, as described by Equations 4 and 5. Please see the newly added visualisations of the diffusion process on our anonymised webpage at https://sites.google.com/view/instant-policy/.
>
> **Regarding the action nodes and alternative approaches.**
>
> That is an excellent point and a good suggestion for the discussion in the related work. For completeness, we have included the work suggested by the reviewer in the related work section.
>
> Our approach can be viewed as optimising the deformation of the graph, but we deform parts of the graph representing actions and do so using a learnt diffusion process, rather than analytically. There are several reasons for this. First of all, we represent robot actions as displacements from the current configuration (i.e. velocities) rather than the desired robot configurations, allowing us to directly execute the predicted actions which results in reactive and robust manipulation policies. Therefore, to represent this relative movement, we include both, the current observation and the actions in the graph representation. Second, our context is composed of robot trajectories rather than a single desired pose. As the reviewer suggests, this problem could be tackled by reaching relative spatial relationships in sequence. However, it would require a module for understanding what the next goal state in a trajectory should be or rely on open-loop execution. Instead, we allow the model to learn at what stage of the task it currently is and what points in the trajectory are relevant (please see the newly added visualisation in our anonymised webpage). Furthermore, our approach allows us to easily integrate multiple demonstrations into the context, enabling generalisation to novel object geometries.
>
> Thus to answer the original question of why action nodes are needed: they express low-level robot actions instead of specific desired key poses, they allow us to simultaneously reason about and predict multiple actions, and circumvent the need for manually identifying the stage of the task (as it is done by nodes representing the current observation).
>
> **Regarding the effort needed to generate pseudo-demonstrations.**
>
> Generating pseudo-demonstrations is an extremely cheap and fast process when compared to collecting the demonstrations in the real world or handcrafting tasks in simulation. Play data that the reviewer refers to is easier to obtain than purposely collecting demonstrations. However, it still requires significant human time and effort, even though it might not be as mundane as doing the same task over and over again to collect a dataset of demonstrations. Additionally, utilising play data to train an In-Cotext model is not straightforward, as we need multiple trajectories performing semantically the same task. Pseudo-demonstrations, on the other hand, can be generated procedurally without much human effort and they do not need to be dynamically feasible as during deployment proper demonstrations will be used to define real tasks. Thus, even computational requirements are substantially smaller due to not needing to run physics simulations.

---

> > ### Author Response · Authors · 2024-11-17
> > **Response to Reviewer ibtP (continued)**
> >
> > **Regarding generalisation to novel objects within the same semantic category.**
> >
> > Although we represent the dense point clouds as a set of nodes, these nodes are equipped with features describing the local geometry around the sampled point. We pre-train an occupancy network that encodes and reconstructs these local geometries, enforcing that the information about the complete object geometry is present in our graph representation (we describe this occupancy network in Appendix A).
> >
> > Thus our model can reason about different object geometries in the context and generalise to completely novel objects at test time. We showcase this in the newly added videos on our anonymous webpage. None of the objects used in these experiments were used during training. Also, note that this is an emergent property of our network - we did not train the model with different objects of the same semantic category. By purposely including such data, the generalisation capabilities could be further increased.
> >
> > **How are sub-goals formulated in the diffusion process? Is it just some horizon the diffusion model is able to generate "in one go"?**
> >
> > Our model predicts a fixed sequence of future actions (in our implementation T=8) expressed as displacements of the end-effector which, when time-scaled, represent end-effector velocities. Our graph representation, in general, can handle an arbitrary number of future action predictions, although with an additional computational cost. It has become a common practice in robot learning to predict multiple actions (~1 second into the future), as it was shown to drastically improve the performance of the learnt policies [1, 2].
> >
> > [1] Zhao, T. Z., Kumar, V., Levine, S., & Finn, C. (2023). Learning fine-grained bimanual manipulation with low-cost hardware. arXiv preprint arXiv:2304.13705.
> > [2] Chi, C., Xu, Z., Feng, S., Cousineau, E., Du, Y., Burchfiel, B., ... & Song, S. (2023). Diffusion policy: Visuomotor policy learning via action diffusion. The International Journal of Robotics Research, 02783649241273668.
> >
> > **How many pseudo-demonstrations are used to train the initial model across how many objects?**
> >
> > We do not generate a fixed number of pseudo-demonstrations but rather keep continuously generating data during training. This approach drastically reduces the data storage requirements and ensures that the model keeps seeing different data every time, preventing overfitting. We utilise the ShapNet dataset, which consists of 12,000 object models, and randomly choose objects from it for each set of pseudo-demonstrations.
> >
> > **Transferring the policy from simulation to the real robot seemed to have required only 25 examples (5 rollouts across 5 tasks). How was performance assessed, and would more samples result in significantly better results?**
> >
> > While, as the reviewer mentions, we used 25 demonstrations collected in the real world to further train our model before evaluating it, we would like to mention that it was able to complete various tasks without incorporating such data, albeit at a lower success rate. We chose to incorporate a small amount of real-world data, because initially we observed that all models, including baselines, often performed poorly due to imperfect segmentations and noisier point clouds than those present in simulation. Thus the main purpose of using the real-world data was not to adapt the model to work on specific types of tasks, but rather to more robustly handle noisy observations of the real world.
> >
> > Our results in simulation showed that adding more realistic, in-domain data can significantly increase the performance on unseen tasks. So to answer the question, we would expect the performance to increase with more samples added, as you suggest. This opens a future work possibility in utilising existing robotics datasets in combination with pseudo-demonstrations, a direction which we are excited about.

---

> ### Comment · Reviewer_ibtP · 2024-11-22
>
> Thank you for your in-depth response to my initial review. I have taken a look at it as well as the reviews and responses from the other reviewers.
>
> Please incorporate some intuition regarding the action nodes into the final paper. I am convinced by your argument about pseudo-demonstration and novel objects, so I have increased my score to accept.

---

> > ### Author Response · Authors · 2024-11-25
> > **Response to Reviewer ibtP**
> >
> > Thank you for your insightful questions and recommendations on how to improve our paper! We believe that your participation in the review process allowed us to greatly improve the quality and clarity of our work. As per your suggestion, we added a discussion regarding the action nodes.
> >
> > We believe pseudo-demonstration can allow us to tackle In-Context Imitation Learning in a scalable and efficient way, and are happy to see that you see the potential of such an approach as well!

---

### Official Review · Reviewer_WsTz · 2024-11-03

**Soundness:** 3
**Presentation:** 3
**Contribution:** 3
**Rating:** 8
**Confidence:** 3

**Summary:**

The paper presents Instant Policy, a method aimed at improving In-Context Imitation Learning (ICIL) by enabling task learning from just one or two demonstrations without additional training. It introduces a graph-based representation to effectively combine demonstrations and observations, framing ICIL as a diffusion-based graph generation problem. The authors also explore training with procedurally generated pseudo-demonstrations, which allows for scalable data generation. The results suggest that this approach can perform well across various tasks in both simulation and real-world contexts, showing potential for scalability and generalization.

**Strengths:**

1. The main contribution of this work is the Instant Policy framework for In-Context Imitation Learning, which enables immediate robotic skill acquisition from just one or two demonstrations using a novel graph diffusion process.
2. The motivation is well-founded. Large Language Models (LLMs) demonstrate impressive capabilities in in-context learning. The paper analyzes the underlying reasons behind the emergence of these capabilities. It also addresses the challenges of acquiring such abilities in the field of robotics and proposes solutions for each challenge identified.
3. The experimental results and demonstrations indicate that the model possesses certain in-context learning capabilities. It also showcases interesting abilities in downstream applications, including the capacity to learn from demonstrations even after cross-embodiment transfer and modality change during the reasoning phase.

**Weaknesses:**

1. The writing in the paper could be improved, particularly in the model training section, which is somewhat unclear. Figure 3's left image does not effectively illustrate this process.
2. The collection and use of pseudo-demonstrations are not well explained, especially regarding how the model utilizes them and the differences in their usage compared to the RLBench dataset or real-world datasets. For details, see the question section.

**Questions:**

1. About baselines: Do the baseline models also learn from pseudo-demonstrations during the training phase? In the inference phase, is the accessed context modality video, images, or a data structure similar to INSTANT POLICY (e.g., point clouds, points)?
2. About training dataset: What types of data were used to train the models tested in simulation versus those tested in the real world? Did both use pseudo-demonstrations generated from ShapeNet?
3. About related works: What is the main difference between your work and One-Shot / Few-Shot Imitation Learning approaches, such as [1]?
4. About inference: I noticed that the model predicts T=8 future actions, and I'm curious whether the model completes the task based on the provided demonstration by making just one prediction.

[1] Zhang, Xinyu, and Abdeslam Boularias. "One-Shot Imitation Learning with Invariance Matching for Robotic Manipulation." arXiv preprint arXiv:2405.13178 (2024).

---

> ### Author Response · Authors · 2024-11-17
> **Response to Reviewer WsTz**
>
> **Regarding the description of the training procedure.**
>
> Thank you for pointing out how our paper could be improved and made easier to understand. The training process can be mainly understood by looking at Equations 1 and 2. In short, training includes 4 main steps: 1) noise is added to the ground truth actions, 2) noisy actions are used to construct our graph representation, 3) the network predicts how nodes representing robot actions need to be adjusted to effectively remove the added noise, and 4) the prediction and ground truth labels are used to calculate the loss function, and weights of the network are updated accordingly. We have added a section in the appendix, describing the training procedure and included an additional figure that more clearly illustrates the training process.
>
> **Regarding how the model utilises pseudo-demonstrations and how they differ from other sources of data.**
>
> Pseudo-demonstrations serve as a primary source of training data for our model. We generate sets of demonstrations performing the same type of pseudo-task (e.g. 3 pseudo-demonstrations for each pseudo-task). Then, to train our model, we use a subset of these demonstrations (e.g. 2 out of 3 demonstrations) to define the context and predict actions from the pseudo-demonstration that was left out when defining the context. Pseudo-demonstrations are never used during inference to define the context.
>
> RLBench and real-world data, on the other hand, can be used either as an additional source of training data or to define the context at inference. For training, it follows the same format as the pseudo-demonstrations. A few demonstrations are collected performing the same task, and some are used to define the context while inferring actions from the other. When used during inference, they are solely used to define the task and the context. We have updated the explanation regarding the use of pseudo-demonstrations and how they differ from other sources of data in the updated paper.
>
> **Do the baseline models also learn from pseudo-demonstrations during the training phase? In the inference phase, is the accessed context modality video, images, or a data structure similar to INSTANT POLICY (e.g., point clouds, points)?**
>
> We adapted all the baselines to operate on point cloud observations using the same pre-trained encoder used for Instant Policy, to have the same number of trainable parameters as our model, and trained them on the same data (removing parts of the baselines that depend on language annotated data, as pseudo-demonstrations do not have such information). We did this to ensure a fair comparison between the methods, avoiding using different modalities and data sources. We indicate this by adding an asterisk to the baselines and describing it on line 325 in the original paper and line 328 in the updated one.
>
> **What types of data were used to train the models tested in simulation versus those tested in the real world? Did both use pseudo-demonstrations generated from ShapeNet?**
>
> Thank you for pointing out a part of the paper that was not initially clear. All models were trained primarily using pseudo-demonstrations. In simulation, we tested two versions of the models: one trained using only pseudo-demonstrations and another trained with a combination of pseudo-demonstrations and a small amount of data from RLBench tasks. In the real-world setting, the model trained on both pseudo-demonstrations and RLBench data was further fine-tuned by adding a small amount of real-world data (five demonstrations for five tasks, none of which were used during evaluation).
>
> In summary, pseudo-demonstrations allow us to generate arbitrary amounts of training data, while incorporating small amounts of in-domain data (e.g., from RLBench or the real world) improves model performance by adjusting to, for example, imperfect segmentations and noisy observations.
>
> **What is the main difference between your work and One-Shot / Few-Shot Imitation Learning approaches, such as [1]?**
>
> While works, such as the one reviewer points out, rely on some sort of formulation of one or few-shot imitation learning as a pose estimation problem, our work directly reasons about the demonstration trajectories and predicts low-level robot actions, such as end-effector velocities. In this way, we allow the model to reason about what stage of a task it is at and what the appropriate actions should be, resulting in reactive and robust manipulation policies (please see newly added videos on our anonymised webpage at https://sites.google.com/view/instant-policy/). Additionally, our method allows us to easily integrate additional demonstration trajectories, enabling generalisation to novel object geometries. For completeness, we added the work the reviewer suggested to the related work section.

---

> > ### Author Response · Authors · 2024-11-17
> > **Response to Reviewer WsTz (continued)**
> >
> > **I noticed that the model predicts T=8 future actions, and I'm curious whether the model completes the task based on the provided demonstration by making just one prediction.**
> >
> > Predicting multiple actions in the future has become a common practice in robot learning. In this work, actions are represented as small relative movements of the end-effector (as explained in Section 3.1). Executing the T=8 predicted action would result in ~1 second of robot motion. In our real-world experiments, we continuously make action predictions in a close-loop fashion, resulting in reactive and robust manipulation policies. Please see the newly added video on our anonymised webpage, where we show this robustness by introducing disturbances during policy rollout.

---

> > > ### Comment · Reviewer_WsTz · 2024-11-23
> > >
> > > Thanks to the authors for their response. In general, my concerns were addressed. I am happy to raise my score.

---

> > > > ### Author Response · Authors · 2024-11-25
> > > > **Response to Reviewer WsTz**
> > > >
> > > > Thank you for your constructive feedback and willingness to increase your score! We are happy to see that you found our approach interesting and importantly promising. We believe that due to your engagement and insightful questions, we managed to improve the quality of our paper.

---

### Official Review · Reviewer_LPk7 · 2024-11-04

**Soundness:** 4
**Presentation:** 4
**Contribution:** 3
**Rating:** 8
**Confidence:** 4

**Summary:**

This paper introduces the Instant Policy method for in-context imitation learning. Instant Policy represents the state as a graph computed from a point cloud and object segmentation. Edges in this graph connect gripper nodes from the current observation to gripper nodes in the in-context demonstrations. Instant Policy generates actions through a graph diffusion process that produces a gripper state in the graph. The model is trained with "pseudo-task" data of a gripper interacting with simulated objects in arbitrary ways. Experiments show that Instant Policy can perform tabletop manipulation tasks from two demonstrations better than ICIL baselines. Instant Policy can also work with cross-embodiment demonstrations and language instructions.

**Strengths:**

1. Instant Policy empirically outperforms ICIL baselines in 12 RLBench tasks by a large margin. This indicates that the graph representation and graph diffusion process from Instant Policy provides meaningful improvements over prior approaches.

1. The pseudo-demonstrations are an effective way to scale data collection for robot manipulation tasks in ICIL. This data source is scalable because it simulates interactions with arbitrary objects without the requirement that they are dynamically or kinematically feasible. The value of this data is confirmed by all methods achieving some non-zero ICIL performance when trained with this pseudo-demonstration source.

1. The architecture of Instant Policy is ablated in Table 2, confirming that the action and diffusion mode are important for performance.

1. The paper shows scaling results for Instant Policy in Figure 6.

1. Instant Policy outperforms baselines on real-world robot experiments. These results show that the Instant Policy assumptions of point cloud and object segmentation hold for real-world scenarios where these sensors will be noisy.

1. The paper shows several interesting extensions of Instant Policy. Table 4 shows Instant Policy generalizes to new objects in the real-world. The graph state representation of Instant Policy allows it to work with human demonstrations for the context as well. Finally, language instructions can also replace the provided demonstration context to work with a language-specified tasks.

1. Supplementary G provides useful transparency about architecture and training decisions that did not work, which will be useful for researchers in the field.

1. Qualitative results are provided in the supplementary section's analysis of the failure modes, the supplementary website, and in Figure 5.

1. The paper and code provide sufficient details for reproducing the results.

**Weaknesses:**

1. It is unclear how to extend the Instant Policy to partially observable settings. Instant Policy relies on all the objects in the scene being represented via nodes in the local graph. Even with addressing partial observability by appending previous local graphs, Instant Policy still would struggle with object clutter or occluded objects in containers. The graph state representation that enables Instant Policy to perform well in ICIL in simple tabletop tasks could be a hindrance when scaling to more complex task settings.

1. The pseudo-demonstration generation lacks collision avoidance behaviors. As the trajectories are not necessarily feasible and are just with a single object, they demonstrate a simple interaction motion. Such data does not show how to avoid other obstacles, such as reaching around another object or into a tight space. This limitation is also confirmed in the RLBench failure cases in L891, which mention that the robot gripper typically moves in a straight line toward objects. Based on this and the previously mentioned weakness, the Instant Policy method seems limited to tabletop manipulation where the scene is fully observable and the gripper only needs to move in a straight line to interact with objects.

1. Instant Policy has two major components: the graph structure and the graph diffusion process. The baselines do not use this same diffusion process or graph structure. The paper does not investigate which of these components of graph diffusion or local graph structure is more important. A simple next-action prediction architecture operating over the graph structure input, like the GPT2 baseline, may be sufficient for good performance like Instant Policy.

1. The method assumes object segmentation to form the local graph. However, this is not a major weakness since the paper demonstrates this works in the real world with noisy observations, so this assumption is likely possible in many realistic settings.

1. The paper uses a fixed number of demonstration steps for all demonstrations being trained on (L819). While Table 2 shows the method is robust to the demonstration length, to scale to more diverse demonstration scenarios, the method must also handle demonstrations of varying lengths where demonstrations with more complex behavior require more steps.

Minor:
1. L88 mentions that Instant Policy achieves "true generalization to unseen tasks". However, the paper never defines what is meant by true generalization and why prior approaches lack this capability.

1. The definition of the term "pseudo-demonstrations" was delayed until late in the paper in Section 3.4. I suggest adding some high-level explanation of what's meant by "pseudo-demonstrations" earlier in the paper since this term is often referenced earlier in the paper.

**Questions:**

1. Perhaps I missed it, but what does the "*" in Table 1 next to the baseline numbers indicate?

1. Why are orientations of objects in the environment fixed to be within some range (L872)? Why not increase the range of rotations in the pseudo-demonstrations or have the policy generalize to these new rotations?

1. Do the objects in RLBench tasks overlap with the objects used in the pseudo-demonstrations? This is important for assessing the generalization abilities of Instant Policy trained only on the pseudo-demonstrations.

1. What exactly does "plateaus at varying levels" on L430 mean? Does this mean the performance does not exceed the numbers in Table 1, even for larger policy sizes?

---

> ### Author Response · Authors · 2024-11-17
> **Response to Reviewer LPk7**
>
> **Regarding the partially observable settings.**
>
> We agree that sometimes occluded objects make the problem more challenging, and we acknowledged this in the limitations section of the original paper. In our real-world experiments, we observed strong performance even with partial point clouds, which are sometimes severely occluded by the robot's gripper. We hypothesise that this is mainly due to the use of local geometry features, which allow the model to identify specific parts of the objects and act accordingly, even if the whole object is not observed.
>
> As the reviewer suggests, incorporating past observations could help further mitigate the impact of partial observability, which would be a simple extension of our graph representation. Additionally, while we currently rely on segmented point clouds, our framework could readily be extended to use point clouds of the entire scene. Randomising distractor objects in the scene during pseudo-demonstration generation would add robustness against occlusions.  Finally, integrating additional semantic features into the observations would further increase the model's ability to reason about partial observations.
>
> Thus, although partial observability and clutter can cause issues for the current implementation of Instant Policy, the overall framework has the potential to be extended to mitigate these issues.
>
> **Regarding collision avoidance.**
>
> We agree with the reviewer that collision avoidance is not addressed in the current implementation of Instant Policy, and we point it out in the limitations section. First, we would like to note that the movement in a straight line that the reviewer mentions is observed only when the robot is approaching the object, and non-linear motions are performed while completing various tasks, such as opening a box.
>
> The limitation of not accounting for collision avoidance itself, however, is not a limitation of the proposed framework in general, but rather of the specific implementation in our experiments. Our formulation of the problem and the graph representation allows us to use dense point clouds of the whole scene, which would provide sufficient information to reason about collision avoidance. Similarly, pseudo-demonstration generation could then be extended to use motion planning to avoid randomised obstacles when approaching the objects of interest. Finally, although in this work, we express the robot state as the pose of the end-effector, the graph representation can be extended to incorporate different parts of the robot, allowing control of the full configuration of the robot through known robot kinematics.
>
> **Regarding a baseline using the next-action prediction architecture operating over the graph structure input.**
>
> In our ablation experiments, we do study the influence that the diffusion process has. One of the ablation variants of Instant Policy, described in Section 4 ("No Diff" in Table 2), does not use diffusion but rather predicts the next action directly, as the reviewer suggests. This comparison allows us to estimate the added value of using diffusion on graphs rather than relying solely on the graph representation. As shown in Table 2, we see a 29% drop in the success rate, indicating that the diffusion process is indeed a crucial part of the method. We attribute this boost in performance when diffusion is used to two factors. First, the diffusion objective is overall more expressive, enabling capturing complex multi-modal distributions. Second, iteratively adjusting the nodes in the graph representing the actions ensures that observations and actions are expressed in the same graph space, and the model does not need to learn complex and highly non-linear mapping between these spaces. Intuitively, such an approach allows the model to 'imagine' spatial implications of considered actions, as nodes are updated as if the actions were executed.
>
> **Regarding the assumption of using object segmentation.**
>
> While the current implementation of Instant Policy assumes access to segmentations, the overall framework could be easily extended to utilise dense point clouds of the entire scene. And as the reviewer points out, the proposed method already demonstrated good performance in real-world settings using noisy observations.

---

> > ### Author Response · Authors · 2024-11-17
> > **Response to Reviewer LPk7 (continued)**
> >
> > **Regarding handling demonstrations of varying lengths.**
> >
> > In this work, we used a fixed number of demonstration steps because the types of tasks we considered could all be described using it. However, this is an implementation detail, and our graph representation can handle an arbitrary number of demonstrations of varying lengths. Through the attention mechanism, the model can look at the demonstrations in the context, understand what stage of the task it is currently at, and what is important to determine the correct actions (please see newly added visualisations in our updated webpage at https://sites.google.com/view/instant-policy/). Thus, increasing the demonstration length would result in the same overall methodology. Importantly, due to the use of directional edges in our graph representation, memory and computational requirements of Instant Policy scale linearly with an increasing number of demonstrations and their length. Additionally, to facilitate more efficient reasoning about the demonstrations, training could be modified to adjust the number of steps based on the complexity of the demonstrations.
> >
> > **Regarding minor improvements to the writing of the paper.**
> >
> > Thank you for the suggestion on how to improve the writing of the paper, to convey the important aspects of the framework. We have updated the paper, incorporating the provided feedback about the minor weaknesses.
> >
> > **Perhaps I missed it, but what does the "*" in Table 1 next to the baseline numbers indicate?**
> >
> > We explain the use of  "*" on L325. It indicates that we adapted all the baselines to operate on point cloud observations using the same pre-trained encoder used for Instant Policy, so that the baselines have the same number of trainable parameters as our model, and also that we trained them on the same data (removing parts of the baselines that depend on language annotated data, because pseudo-demonstrations do not have such information). We did this to ensure a fair comparison between the methods in order to focus our experiments on answering specific questions about design choices.
> >
> > **Why are orientations of objects in the environment fixed to be within some range (L872)? Why not increase the range of rotations in the pseudo-demonstrations or have the policy generalize to these new rotations?**
> >
> >
> > We restricted the orientation of objects in the environment mainly due to the kinematic limitations of a robotic manipulator used in our simulation experiments. Some tasks in this setting can be completed with arbitrary object orientations, and some become kinematically infeasible because of the robot embodiment. To keep the setting consistent, we, therefore, restricted orientation for all of the tasks.  The reviewer also correctly points out that the range of rotations in the pseudo-demonstrations could be increased. Note that when coupled with a randomised initial gripper pose, this range becomes much larger. Overall, restricted orientation is an implementation detail, and our framework can handle arbitrary object orientations.
> >
> > **Do the objects in RLBench tasks overlap with the objects used in the pseudo-demonstrations?**
> >
> > No object instances used in RLBench tasks were used to generate pseudo-demonstrations. We solely used objects from the ShapeNet dataset to generate the pseudo-demonstrations. It is worth noting that some semantic categories of objects (e.g. laptops) used in RLBench are present in this dataset, although they have different geometries.
> >
> > **What exactly does "plateaus at varying levels" on L430 mean?**
> >
> > Thank you for pointing out the part of the paper that was not well explained. We have updated the paper to convey this message more clearly.
> >
> > In summary, while we see decreasing validation loss with increased model training time, the success rate on unseen RLBench tasks plateaus. Increasing the model's capacity from 69M to 117M, the success rate reached before plateauing increases significantly. However, further increasing the number of trainable parameters to 178M results in only minor, insignificant improvements in performance.
> >
> > However, our simulation results show that incorporating small amounts of more realistic, in-domain data can drastically boost the performances. This opens a future work possibility in utilising existing robotics datasets in combination with pseudo-demonstrations, a direction which we are excited about. We hypothesise that in such a setting, scaling capacity of the model would increase the overall performance even further.

---

> ### Comment · Reviewer_LPk7 · 2024-11-23
>
> Thank you for the response. The response sufficiently addresses my listed weaknesses, so I raise my score to accept.

---

> > ### Author Response · Authors · 2024-11-25
> > **Response to Reviewer LPk7**
> >
> > Thank you for participating in the review process and helping us improve the quality of our paper! We appreciate your willingness to increase your score, given the improvements to the paper. We are excited about the potential of this work and are happy to see you found it interesting.

---

### Author Response · Authors · 2024-11-17
**General Response**

Thank you very much to all reviewers for your comments and suggestions on how to improve our paper and insightful questions. We are pleased to see that the reviewers found our work interesting and our experiments convincing. We have written our responses individually to each reviewer, addressing their concerns separately. We have also uploaded our updated paper with all new material in blue text. To see our new visualisations and robot videos, please visit our updated anonymised webpage at https://sites.google.com/view/instant-policy/.

If reviewers have any further questions, please feel free to ask—we would be very happy to discuss them.

---

### Author Response · Authors · 2024-12-04
**Closing Remarks**

As the rebuttal period ends, we would like to thank the reviewers once again for spending their time reviewing our work and helping us to improve it.

The reviewers raised insightful questions on key aspects such as data usage, action representation, training procedure, limitations acknowledged in the paper and presented results. We have answered these questions by providing additional details, refining our explanations, and incorporating suggested improvements into our paper. The rebuttal period and the reviewers' suggestions allowed us to significantly improve the quality and clarity of our paper. We have uploaded the revised version of the paper.

We are really excited about the promise of Instant Policy and are glad to see that the reviewers also see its potential. It already showed the ability to perform various unseen manipulation tasks instantly after just one or a few demonstrations, unlocking new possibilities for In-Context Imitation Learning. Showing that such results can be achieved using mostly cheap procedurally generated data in simulation is a strong indicator that current limitations in robotics due to data scarcity could now be overcome by scaling up this simulated data.

---

### Meta-Review · Area_Chair_avjb · 2024-12-11

**Metareview:**

The proposed Instant Policy is a method for in-context imitation learning (ICIL) that lets robots learn tasks from just one or two demonstrations with a pre-trained model, without needing task-specific fine-tuning. The approach uses a graph-based representation of demonstrations and observations, framing ICIL as a diffusion-based graph generation process. It’s trained on procedurally generated pseudo-demonstrations, which helps provide scalable and diverse data for learning. Experiments show strong performance on tabletop manipulation tasks, with promising results in both simulation and some real-world evaluations.

Almost unanimously, the reviewers appreciated the introduction of Instant Policy for in-context imitation learning, especially its graph-based representation and the use of scalable, procedurally generated data for pretraining. Several reviewers pointed out the method’s potential versatility in handling cross-embodiment demonstrations and modality changes. Reviewers also asked questions about the design choices of the method, how pseudo-demonstrations were generated, action representation, how the work positions itself relative to other subareas, the diffusion process, and writing clarity.

Through the rebuttal, most of the raised concerns were addressed. Based on this, three reviewers shifted from weak to clear acceptance. The AC agrees with the strong and unanimous recommendation of the committee to accept this paper.

**Additional Comments On Reviewer Discussion:**

- Three reviewers (`LPk7`, `WsTz`, `ibtP`) increased their scores from an original rating of 6 to a final rating of 8.
- Reviewer `Wb4x` maintained their original score of 8.

All reviewers acknowledged the authors' responses and stated that their concerns were adequately addressed.

---

### Decision · Program_Chairs · 2025-01-22

Accept (Oral)